# Supervised Disentanglement Under Hidden Correlations

## Abstract

Disentangled representation learning (DRL) is a powerful paradigm for improving the generalization of representations. While recent DRL methods attempt to handle attribute correlations by enforcing conditional independence based on attributes, they overlook the realities of complex multi-modal data distributions and hidden correlations under attributes. We theoretically show that, under such hidden correlations, existing methods lose mode information and fail to achieve disentanglement. To address this gap, we introduce Supervised Disentanglement under Hidden Correlations (SD-HC), a framework that explicitly discovers data modes under attributes and minimizes mode-based conditional mutual information. Theoretically, we establish that SD-HC provides sufficient conditions for disentanglement in the presence of hidden correlations, preserving mode and attribute information. Empirically, SD-HC shows improved generalization compared to the state-of-the-art baselines across toy data and seven real-world datasets. Code is available at `https://anonymous.4open.science/r/SD-HC-1FAD`.

## 1 Introduction

Disentangled representation learning (DRL) aims to encode each data attribute in its corresponding representation subspace, which holds great promise in enhancing generalization to unseen scenarios (Matthes et al., 2023; Qian et al., 2021), enabling controllable generative modeling (Yuan et al., 2021), and improving fairness (Locatello et al., 2019a). In the supervised setting, each representation subspace is learned under the label supervision of its corresponding attribute, while being disentangled from other attributes.

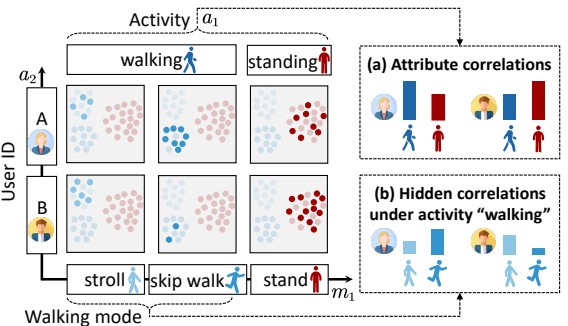

Figure 1: Correlated human activity data. The distributions of (a) **"walking" / "standing"** and (b) **"stroll" / "skip walk"** under "walking" differ between users, exhibiting correlations.

Supervised DRL methods typically assume independence between attributes. In addition to supervised prediction, mutual information (MI) minimization (Kwon et al., 2020; Yuan et al., 2021; Su et al., 2022) is commonly adopted to achieve disentanglement by enforcing independence between the representations of different attributes. The independence assumption is often violated in real-world data, where correlations are prevalent. Taking human activities as an example, different users have different behavior patterns, and each user tends to engage in some activities more frequently than others, exhibiting correlations between activity and user identity (ID) attributes, as shown in Figure 1(a). For correlated attributes, enforcing representation independence causes at least one subspace to lose attribute information (Funke et al., 2022).

To disentangle correlated attributes, attribute-based conditional mutual information minimization (A-CMI) (Funke et al., 2022) enforces conditional representation independence that preserves attribute information. However, when a certain attribute takes a value, underlying variations related to this attribute may lead to a complex *multi-modal* data distribution, characterized by multiple

high-density regions, each referred to as a *mode*. The mode under this value of this attribute may be correlated with other attributes. Continuing with the human activity example, when activity attribute takes the value "walking", variations in pace, stride, and posture may lead to different walking modes, the relaxed "stroll" and energetic "skip walk"; different users have more subtle differences in their behavior patterns, exhibiting correlations between walking mode and user ID attribute, as shown in Figure 1(b). In this case, A-CMI may cause the loss of mode information (as proved in Proposition 1), which is important for attribute prediction (Nie et al., 2020; Sugiyama, 2021; Li et al., 2017). The modes under different attribute values may form adjacent or interleaved cluster structures, where preserving such local structures benefits attribute prediction, e.g., the "skip walk" mode of walking resembles the activity "climbing down", and explicitly encoding this easily confused mode helps to recognize the "walking" activity.

To address the above problem, we propose Supervised Disentanglement under Hidden Correlations (SD-HC). Instead of focusing on attribute correlations as existing works, we delve into the complex data distributions and hidden correlations under certain attributes. Our contributions are:

- We prove that mode-based CMI minimization is the *necessary and sufficient condition* for supervised disentanglement under hidden correlations and attribute correlations. While existing works have not established the sufficient condition for disentanglement under correlations, we show that CMI minimization can achieve disentanglement under various correlation types, establishing the first *sufficient condition*.

- We introduce a novel supervised DRL method under hidden correlations, SD-HC, designed as a model-agnostic framework that implements the sufficient conditions for disentanglement based on discovered data modes. By minimizing mode-based CMI, SD-HC disentangles attributes while preserving mode information that existing methods tend to lose.

- We extensively evaluate SD-HC on toy data and seven real-world datasets, demonstrating the superiority of SD-HC in attribute prediction tasks across distribution shifts and train-test correlation shifts. Comprehensive investigations validate the generalization ability and predictive ability of the learned representations.

## 2 RELATED WORK

**Disentanged Representation Learning.** DRL methods can be roughly divided into unsupervised, weakly-supervised, and supervised DRL. Unsupervised DRL learns independent representation dimensions that each correspond to an unknown attribute by self-supervision, e.g., variational auto-encoding (Higgins et al., 2016; Kim & Mnih, 2018; Chen et al., 2018) or contrastive learning (Zimmermann et al., 2021; Matthes et al., 2023). Yet, the feasibility of purely unsupervised disentanglement has been questioned (Locatello et al., 2019b), which prompts DRL with weak supervision (Shu et al., 2020), e.g., similarity (Chen & Batmanghelich, 2020) or grouping information (Bouchacourt et al., 2018). Supervised DRL learns one multi-dimensional representation subspace for each labeled attribute (Qian et al., 2021; Yuan et al., 2021). Generally, DRL methods assume attribute independence and enforce representation independence between different attributes for disentanglement. We study supervised DRL, which usually minimizes the MI between representations (Kwon et al., 2020; Yuan et al., 2021; Su et al., 2022), minimizes the Maximum Mean Discrepancy (MMD) between representation distributions (Li et al., 2018; Lin et al., 2020), or makes one attribute unpredictable from the representations of another by adversarial training (Qian et al., 2021; Li et al., 2022; Lee et al., 2021).

**Disentanglement Under Attribute Correlations.** Recent works show that independence constraints fail to disentangle correlated attributes, causing entanglement for unsupervised DRL (i.e., one dimension encodes several correlated attributes) (Träuble et al., 2021) or hurting the predictive ability of representations for supervised DRL (Funke et al., 2022). To disentangle correlated attributes for unsupervised DRL, adding weak supervision could correct the model (Träuble et al., 2021; Dittadi et al., 2021); using Hausdorff distance can relax independence constraints to encourage factorized supports instead of factorized distributions (Wang & Jordan, 2024; Roth et al., 2023). These methods can somewhat alleviate entanglement but do not guarantee disentanglement theoretically (Funke et al., 2022; Wang & Jordan, 2024).

More recently, conditional independence constraints have been introduced to disentangle correlated attributes. For supervised DRL, A-CMI (Funke et al., 2022) minimizes the CMI based on each attribute between its representation and the joint representations of other attributes, and proves this to be the *necessary* condition for disentanglement. For DRL in reinforcement learning (RL), CMID (Dunion et al., 2023) assumes RL agents act in a temporal Markov Decision Process, and minimizes the CMI based on observed action-representation histories to bypass unobserved current state features.

To the best of our knowledge, existing works have only established *necessary* conditions for DRL under correlations (Wang & Jordan, 2024; Funke et al., 2022), and the sufficiency of CMI minimization has only been validated on linear regression examples without formal proofs. We give the first *sufficient* conditions for DRL under correlations based on CMI, which hold for multiple attributes and varying types of correlations.

# 3 DISENTANGLING UNDER HIDDEN CORRELATIONS

## 3.1 PROBLEM FORMULATION

**Data Generation Process.** We assume data are generated according to the causal process in Definition 1 (Figure 2) based on three key assumptions as listed below. The first is a standard assumption in DRL that must strictly hold (Suter et al., 2019; Wang & Jordan, 2024), while the others are specific to our method but can be relaxed, as discussed near the end of Section 3.3.

**Definition 1.** (Disentangled Causal Process). *Consider a causal generative model $p(\boldsymbol{x}|\boldsymbol{a})$ for data $\boldsymbol{x}$ with $K$ attributes $\boldsymbol{a} = (a_1, a_2, ..., a_K)$. A certain attribute $a_k$ is associated with a categorical mode variable $m_k$. Attributes $\boldsymbol{a}$ are influenced by $L$ confounders $\boldsymbol{c}^{\mathrm{a}} = (c_1^{\mathrm{a}}, ..., c_L^{\mathrm{a}})$. Conditioned on $a_k$, mode variable $m_k$ and other attributes $\boldsymbol{a}_{-k}$ are influenced by $Q$ confounders $\boldsymbol{c}^{\mathrm{m}} = (c_1^{\mathrm{m}}, ..., c_Q^{\mathrm{m}})$. This causal model is called disentangled if and only if it follows a structural causal model (SCM) (Pearl, 2009) of the form:*

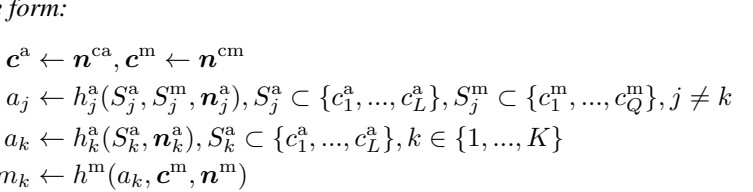

$$
\begin{aligned}
&\boldsymbol{c}^{\mathrm{a}} \leftarrow \boldsymbol{n}^{\mathrm{ca}}, \boldsymbol{c}^{\mathrm{m}} \leftarrow \boldsymbol{n}^{\mathrm{cm}} \\
&a_j \leftarrow h_j^{\mathrm{a}}(S_j^{\mathrm{a}}, S_j^{\mathrm{m}}, \boldsymbol{n}_j^{\mathrm{a}}), S_j^{\mathrm{a}} \subset \{c_1^{\mathrm{a}}, ..., c_L^{\mathrm{a}}\}, S_j^{\mathrm{m}} \subset \{c_1^{\mathrm{m}}, ..., c_Q^{\mathrm{m}}\}, j \neq k \\
&a_k \leftarrow h_k^{\mathrm{a}}(S_k^{\mathrm{a}}, \boldsymbol{n}_k^{\mathrm{a}}), S_k^{\mathrm{a}} \subset \{c_1^{\mathrm{a}}, ..., c_L^{\mathrm{a}}\}, k \in \{1, ..., K\} \\
&m_k \leftarrow h^{\mathrm{m}}(a_k, \boldsymbol{c}^{\mathrm{m}}, \boldsymbol{n}^{\mathrm{m}}) \\
&\boldsymbol{x} \leftarrow g(\boldsymbol{a}_{-k}, m_k, \boldsymbol{n}^{\mathrm{x}})
\end{aligned}
$$

$$(1)$$

Figure 2: Causal graph of data generation under hidden correlations regarding a certain $a_k$.

*with functions $g$, $h_i^{\mathrm{a}}$, $h^{\mathrm{m}}$, jointly independent noises $\boldsymbol{n}^{\mathrm{ca}}$, $\boldsymbol{n}^{\mathrm{cm}}$, $\boldsymbol{n}_i^{\mathrm{a}}$, $\boldsymbol{n}^{\mathrm{m}}$, $\boldsymbol{n}^{\mathrm{x}}$, and confounder subsets $S_i^{\mathrm{a}}$, $S_j^{\mathrm{m}}$, for $i = 1, ..., K, j = 1, ..., K, j \neq k$. $-k$ denotes the set of attribute indices $\{j\}_{j \neq k}$.*

**Key Assumptions.** ❶ Each attribute is an *elementary ingredient* that has no causal effect on other attributes (Suter et al., 2019), i.e., interventions on one attribute do not influence others. ❷ For some value $\alpha$ of attribute $a_k$, $p(\boldsymbol{x}|a_k = \alpha)$ might be a *multi-modal distribution*, e.g., a Gaussian mixture. Each high-density region of the distribution corresponds to a cluster and is referred to as a *mode*. Modes are indexed sequentially by attribute value (e.g., $0 \sim 2$ for $\alpha = 0$, $3 \sim 5$ for $\alpha = 1$), and a categorical mode label $m_k$ is assigned to each sample. ❸ Correlations may arise from two confounder sets: $\boldsymbol{c}^{\mathrm{a}}$ induces *attribute correlations* $I(a_i; a_{i'}), i \neq i'$; $\boldsymbol{c}^{\mathrm{m}}$ induces *hidden correlations* $I(m_k; a_{-k}|a_k) = \sum_\alpha p_{a_k}(a_k = \alpha)I(m_k; a_{-k}|a_k = \alpha)$, i.e., the expectation of the correlation between the modes under $a_k = \alpha$ and other attributes $a_{-k}$. $I(\cdot; \cdot)$ denotes mutual information.

## 3.2 THE DEFINITIONS OF DISENTANGLED REPRESENTATIONS

The goal of supervised DRL is to learn disentangled representations $\boldsymbol{z}_i$ for each labeled attribute $a_i$ by a mapping $f(\boldsymbol{x}) = (\boldsymbol{z}_i)_{i=1}^K, \boldsymbol{z}_i \in \mathbb{R}^D$. Disentangled $\boldsymbol{z}_i$ should (1) contain all information about $a_i$ (**Informativeness**), including any mode information, i.e., $I(\boldsymbol{z}_i; a_i) = H(a_i)$ and $I(\boldsymbol{z}_i; m_i) = H(m_i)$, and (2) respect the causal generative structure by remaining invariant to interventions on another attribute $a_j, j \neq i$ (**Independence**), as in Definition 2 following (Suter et al., 2019).

**Definition 2.** (Disentangled Representation). *Representation $\boldsymbol{z}$ is disentangled, if for $i = 1, ..., K$:*

$$p(\boldsymbol{z}_i|\mathrm{do}(a_{-i})) = p(\boldsymbol{z}_i) \tag{2}$$

where $a_{-i}$ indicates the joint variable of $\{a_j\}_{j \neq i}$, and $\mathrm{do}(a_{-i})$ assigns values to $a_{-i}$ by external intervention outside the causal process and leaves $a_i$ unchanged. Equation 2 requires that $\boldsymbol{z}_i$ depends solely on $a_i$ and is unaffected by changes in other attributes, reflecting post-interventional invariance.

### 3.3 Theoretical Guarantees for Disentangling with Mode-Based CMI Minimization

We focus on the DRL of a certain attribute $a_k$ with underlying modes. For simplicity, we take $K = 2, k = 1$ as an example. The causal graph of representation learning is shown in Figure 3c. While this causal structure remain fixed, different learning objectives make $z_k$ encode different information from data, resulting in varying distributions of $z_k$. We prove that under the data generation process of Definition 1, A-CMI fails under hidden correlations, while mode-based CMI minimization is the necessary and sufficient condition for supervised disentanglement under various correlations. Finally, our results are generalized to multiple attributes and simple cases.

**A-CMI Fails Under Hidden Correlations.** We show that enforcing attribute-based conditional independence (A-CMI), $I(\boldsymbol{z}_1; \boldsymbol{z}_2|a_1) = 0$, could hurt the predictive ability of representations, which is formalized in Proposition 1 and proved in Appendix B.2.

**Proposition 1.** *If $I(m_1; a_2|a_1) > 0$, then enforcing $I(\boldsymbol{z}_1; \boldsymbol{z}_2|a_1) = 0$ leads to at least one of $I(\boldsymbol{z}_1; m_1) < H(m_1)$ and $I(\boldsymbol{z}_2; a_2) < H(a_2)$.*

where $H(\cdot)$ denotes entropy, and the MI $I(\cdot \, ; \cdot)$ between a representation and an attribute measures the amount of information the representation contains about the attribute. $I(\boldsymbol{z}_1; m_1) < H(m_1)$ indicates that $\boldsymbol{z}_1$ loses mode information about $m_1$, which is important for predicting $a_1$, while $I(\boldsymbol{z}_2; a_2) < H(a_2)$ indicates that $\boldsymbol{z}_2$ loses attribute information for predicting $a_2$. Thus, minimizing attribute-based CMI hurts the predictive ability of representations under hidden correlations. This is an extension of Proposition 3.1 in (Funke et al., 2022), which proves that unconditional MI minimization fails under attribute correlations.

**The Necessary Condition for Disentanglement.** A proper independence constraint should be a necessary condition for disentanglement, preserving the predictive ability of representations (***Informativeness***). To identify such constraint, we turn to the properties of the true *latent* representations $\boldsymbol{z}_i^1, i = 1, 2$. For example, on human activity data with activity attribute $a_1$, $\boldsymbol{z}_1^1$ encodes the body movements that characterize activities, which are unaffected by changes in user behavior patterns.

Based on Definition 1, we build the causal graphs of data generation with $\boldsymbol{z}_i^1, i = 1, 2$. Since the disentangled $\boldsymbol{z}_i$ aims to recover the true latent $\boldsymbol{z}_i^1$ and retain its properties, we derive conditional independence between the true *latent* representations as a necessary condition for disentanglement. As stated by the causal graph theorems in Appendix D.1, two variables $X, Y$ are conditionally independent given a variable that blocks all *backdoor paths* between them, i.e., the paths that flow backward from $X$ or $Y$. In Figure 3(a), we consider only attribute correlations as A-CMI, where $a_1$ blocks the only *backdoor path* between $\boldsymbol{z}_1^1$ and $\boldsymbol{z}_2^1$. In comparison, we consider additional hidden correlations in Figure 3(b), where $m_1$

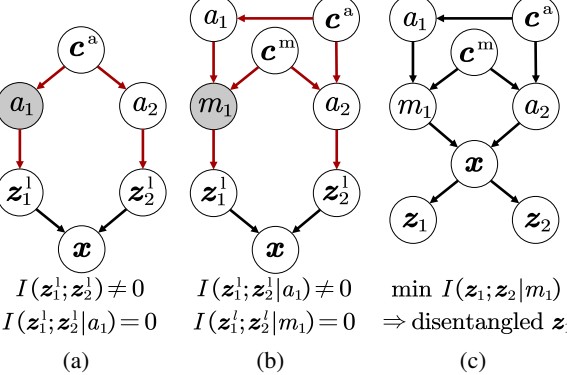

$$I(\boldsymbol{z}_1^1; \boldsymbol{z}_2^1) \neq 0 \qquad I(\boldsymbol{z}_1^1; \boldsymbol{z}_2^1|a_1) \neq 0 \qquad \min I(\boldsymbol{z}_1; \boldsymbol{z}_2|m_1)$$
$$I(\boldsymbol{z}_1^1; \boldsymbol{z}_2^1|a_1) = 0 \qquad I(\boldsymbol{z}_1^1; \boldsymbol{z}_2^1|m_1) = 0 \qquad \Rightarrow \text{disentangled } \boldsymbol{z}_1$$
(a) \qquad\qquad (b) \qquad\qquad (c)

Figure 3: Causal graphs of representations. (a) and (b): Data generation with the true *latent* representations $\boldsymbol{z}_1^1, \boldsymbol{z}_2^1$, where **Red** arrows indicate the *backdoor paths* between them. (c): Representation learning that produces the *learned* representations $\boldsymbol{z}_1, \boldsymbol{z}_2$.

blocks all *backdoor paths*, yet $a_1$ fails to block the path via $\boldsymbol{c}^\mathrm{m}$ (consistent with the failure of A-CMI under hidden correlations). Thus, under hidden correlations and attribute correlations, the true latent representations are conditionally independent based on $m_1$, a property the learned disentangled

representations must retain:

$$I(\boldsymbol{z}_1^1; \boldsymbol{z}_2^1|m_1) = 0 \ \Rightarrow \ \textit{If } \boldsymbol{z}_1 \textit{ is the disentangled representation of } a_1, \textit{ then } I(\boldsymbol{z}_1; \boldsymbol{z}_2|m_1) = 0 \quad (3)$$

**The Sufficient Condition for Disentanglement.** We further show that, assuming data are generated as in Definition 1, mode-based CMI minimization yields ***Independence*** and thus suffices for supervised disentanglement under various correlations, as formalized in Proposition 2.

**Proposition 2.** *Under the data generation process of Definition 1 ($K = 2, k = 1$), if $I(\boldsymbol{z}_1; m_1) = H(m_1)$, $I(\boldsymbol{z}_2; a_2) = H(a_2)$, and $I(\boldsymbol{z}_1; \boldsymbol{z}_2|m_1) = 0$, then $p(\boldsymbol{z}_1|\text{do}(a_2)) = p(\boldsymbol{z}_1)$, i.e., $\boldsymbol{z}_1$ is the disentangled representation of $a_1$.*

As the impact of external interventions cannot be directly evaluated (Wang & Jordan, 2024), we prove Proposition 2 in two steps. First, using mutual information theory, we show that mode-based CMI minimization leads to conditional independence $I(\boldsymbol{z}_1; a_2|m_1) = 0$, limiting the information $\boldsymbol{z}_1$ contains about $a_2$ (Lemma 2.1, proof in Appendix B.3.1). Second, using do-calculus on the causal graph in Figure 3(c), we show that this conditional independence implies post-interventional invariance $p(\boldsymbol{z}_1|\text{do}(a_2)) = p(\boldsymbol{z}_1)$ based on the data generation process of Definition 1 (Lemma 2.2, see Appendix B.3.2). Notably, Lemma 2.1 reveals the validity under attribute correlations and hidden correlations.

**Lemma 2.1.** *If $I(\boldsymbol{z}_1; m_1) = H(m_1)$, $I(\boldsymbol{z}_2; a_2) = H(a_2)$, and $I(\boldsymbol{z}_1; \boldsymbol{z}_2|m_1) = 0$, then $I(\boldsymbol{z}_1; a_2) = I(m_1; a_2)$ and $I(\boldsymbol{z}_1; a_2|m_1) = 0$.*

where $I(m_1; a_2)$ denotes the *total hidden correlations*, decomposing as the sum of attribute correlations and hidden correlations, i.e., $I(m_1; a_2) = I(a_1; a_2) + I(m_1; a_2|a_1)$, as proved in Appendix B.1. Thereby, $I(\boldsymbol{z}_1; a_2) = I(m_1; a_2)$ shows that $\boldsymbol{z}_1$ contains information about $a_2$ only if it is induced by correlations regarding its attribute or mode. Further, $I(\boldsymbol{z}_1; a_2|m_1) = 0$ shows that $\boldsymbol{z}_1$ contains no additional information about $a_2$ knowing its mode.

**Scope of Applicability: Key Assumptions and Generalizations.** Our theoretical results are based on the data generation process of Definition 1, relying on the causal structure (Assumption ❶, attributes as elementary ingredients) and not restricted to specific functional forms or parameterizations. They naturally extend to (1) *multiple attributes* ($K > 2$), where the extension mainly involves replacing single variables $a_2, \boldsymbol{z}_2$ with joint variables $a_{-k}, \boldsymbol{z}_{-k}$ (Equation 7 and Corollaries 2, 2.1, 2.2, see Appendix B.4); (2) *uni-modal distributions with attribute correlations*, where only one mode exists under each attriubute value (Assumption ❷ relaxation), and mode-based CMI degrades to attribute-based CMI; and (3) *uncorrelated data* as correlation strengths vanish (Assumption ❸ relaxation). Although our results strictly rely on the elementary-ingredient assumption, they extend to arbitrary parameterizations, numbers of attributes/modes, and correlation types/strengths.

**Theoretical Contributions.** We prove the sufficiency of CMI minimization for supervised disentanglement, which has only been validated on linear regression examples (Funke et al., 2022). *This is the first attempt to establish sufficient conditions for disentanglement under correlations*, unlike necessary conditions before (Wang & Jordan, 2024; Funke et al., 2022). Our results generalize to multiple attributes and various cases, showing that *one independence constraint is sufficient for the supervised DRL of one attribute*.

# 4 METHOD

**Framework.** We show the framework of SD-HC for disentangling a certain attribute $a_k$ with hidden correlations in Figure 4, which consists of encoder $F$ for learning representations $F(\boldsymbol{x}) = \boldsymbol{z} =$

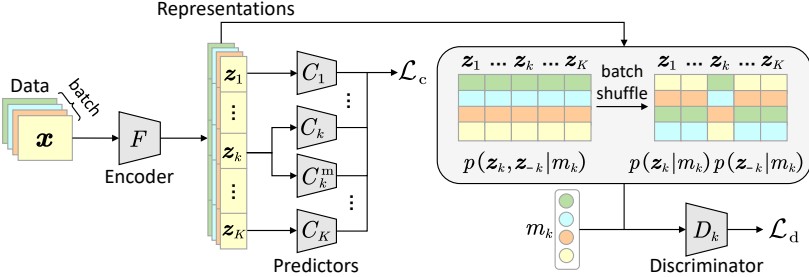

Figure 4: SD-HC Framework for disentangling a certain $a_k, k \in \{1, ..., K\}$ with underlying modes.

$(\boldsymbol{z}_1, \boldsymbol{z}_2, ..., \boldsymbol{z}_K), \boldsymbol{z}_i \in \mathbb{R}^D, i = 1, ..., K$, predictors $\{C_i\}_{i=1}^K$ for predicting each attribute, predictor $C_k^{\mathrm{m}}$ for predicting mode $m_k$, and discriminator $D_k$ for minimizing mode-based CMI. Our method builds on mode labels $m_k$ estimated prior to training (see Section 5.1), which is an external step that can be tailored to the data. SD-HC is architecture-agnostic and can be used in various applications.

The framework can be expanded to disentangle multiple attributes by adding one independence constraint to disentangle each attribute. The form of independence constraints depends on the correlation types, i.e., attribute-based CMI for attribute correlations or mode-based CMI for hidden correlations. For supervised constraints, $I(\boldsymbol{z}_i; a_i) = H(a_i), i = 1, ..., K$ are always required, along with one additional constraint $I(\boldsymbol{z}_i; m_i) = H(m_i)$ for each attribute $a_i$ with underlying modes. For additional constraints, discriminators and mode predictors should be added accordingly.

**Losses.** The losses are strictly designed according to the sufficient conditions for disentanglement in Proposition 2. As commonly done in adversarial training (Chen et al., 2023), optimizations w.r.t. different losses are performed alternatively. See Appendix F for the detailed training process.

(1) To enforce informativeness constraints such as $I(\boldsymbol{z}_k; a_k) = H(a_k)$, since maximizing mutual information between representations and their labels equals minimizing the standard cross-entropy (Boudiaf et al., 2020), we minimize supervised loss $\mathcal{L}_{\mathrm{c}}$ with attribute and mode prediction losses $\mathcal{L}_{\mathrm{ac}}, \mathcal{L}_{\mathrm{mc}}$ as follows:

$$\mathcal{L}_{\mathrm{c}} = \mathcal{L}_{\mathrm{ac}} + w_{\mathrm{m}} \cdot \mathcal{L}_{\mathrm{mc}}, \quad \mathcal{L}_{\mathrm{ac}} = \mathbb{E}_{\boldsymbol{x}}[\textstyle\sum_{i=1}^K l_{\mathrm{ce}}(C_i(\boldsymbol{z}_i), a_i)], \quad \mathcal{L}_{\mathrm{mc}} = \mathbb{E}_{\boldsymbol{x}}[l_{\mathrm{ce}}(C_k^{\mathrm{m}}(\boldsymbol{z}_k), m_k)] \quad (4)$$

where $w_{\mathrm{m}}$ is the weight of mode prediction loss, and $l_{\mathrm{ce}}(\cdot)$ denotes cross entropy function.

(2) To minimize mode-based CMI $I(\boldsymbol{z}_k; \boldsymbol{z}_{-k}|m_k)$, since $I(\boldsymbol{z}_k; \boldsymbol{z}_{-k}|m_k) = 0$ is equivalent to $p(\boldsymbol{z}_k, \boldsymbol{z}_{-k}|m_k) = p(\boldsymbol{z}_k|m_k)p(\boldsymbol{z}_{-k}|m_k)$, we minimize CMI by matching the joint distribution $p(\boldsymbol{z}_k, \boldsymbol{z}_{-k}|m_k)$ and the marginal distribution $p(\boldsymbol{z}_k|m_k)p(\boldsymbol{z}_{-k}|m_k)$ with adversarial training (Belghazi et al., 2018). For mode $\mu$, representations $(\boldsymbol{z}_1, \boldsymbol{z}_2)$ are sampled from their joint and marginal distributions by the following procedure: first, we select data in the mini-batch under $m_k = \mu$; the joint representation pairs are formulated by concatenating $\boldsymbol{z}_1, \boldsymbol{z}_2$ of the same sample, and the marginal representation pairs are formulated by concatenating $\boldsymbol{z}_k$ with $\boldsymbol{z}_{-k}$ jointly shuffled within this mode (Funke et al., 2022; Dunion et al., 2023). Given the sampled representation pairs, Jensen-Shannon Divergence is used to measure the discrepancy between the two distributions for stability (Hjelm et al., 2019). Discrimination loss $\mathcal{L}_{\mathrm{d}}$ is formulated as follows, where $l_{\mathrm{bce}}(\cdot)$ denotes binary cross entropy function:

$$\mathcal{L}_{\mathrm{d}} = \mathbb{E}_{m_k}[\mathbb{E}_{(\boldsymbol{z}_k, \boldsymbol{z}_{-k})|m_k}[l_{\mathrm{bce}}(D(\boldsymbol{z}_k, \boldsymbol{z}_{-k}, m_k), 1)] + \mathbb{E}_{\boldsymbol{z}_k|m_k, \boldsymbol{z}_{-k}|m_k}[l_{\mathrm{bce}}(D(\boldsymbol{z}_k, \boldsymbol{z}_{-k}, m_k), 0)]] \tag{5}$$

## 5 EXPERIMENTS

### 5.1 EXPERIMENTAL SETTINGS

**Datasets.** We evaluate on toy data and seven real-world datasets, namely Colored MNIST (**CMNIST**) constructed from MNIST (Arjovsky et al., 2019), Colored Fashion-MNIST (**CFashion-MNIST**) constructed from Fashion-MNIST (Xiao et al., 2017), Canine-Background (**Canine-BG**) constructed from ImageNet (Deng et al., 2009), **UCI-HAR** (Anguita et al., 2013), **RealWorld** (Sztyler & Stuckenschmidt, 2016), **HHAR** (Stisen et al., 2015), and **MFD** (Lessmeier et al., 2016). We define attributes $a_1, a_2$ as the generative factors of dimensions $\boldsymbol{x}_1, \boldsymbol{x}_2$ on toy data, parity ("even" or "odd") and color of digits on CMNIST, the style ("sporty" or "chic") and color of clothing on CFashion-MNIST, the functional categories and image backgrounds of dogs on Canine-BG, fault type and operating condition on MFD, and activity and user ID on other wearable human activity recognition (WHAR) datasets. The task is to learn disentangled representations for $a_1$ with underlying modes, which correspond to digits and items on CMNIST and CFashion-MNIST, respectively, e.g., digit "2" under parity "even", item "sneaker" under style "sporty", and breed "silky terrier" under functional category "pet". See Appendix G for details.

**Evaluation Protocols.** *On toy data, CMNIST, CFashion-MNIST, and Canine-BG with constructed modes, correlation shifts are introduced by sampling* (Roth et al., 2023). *We train on correlated data and evaluate on 3 test sets with increasing train-test correlation shifts, namely test 1 (same correlations), test 2 (no correlations), and test 3 (anticorrelations).* For CMNIST, CFashion-MNIST,

Table 1: Comparison with baselines (mean±std, in percentage). "*" indicates SD-HC is statistically superior to baselines by pairwise t-test at a 95% significance level. The best and runner-up results are **bold** and underlined, respectively. Improvements by SD-HC are computed over the best baseline.

| Method | CMNIST | | CFashion-MNIST | | Canine-BG | | UCI-HAR | | RealWorld | | HHAR | | MFD | |
|---|---|---|---|---|---|---|---|---|---|---|---|---|---|---|
| | Acc. | Mac. F1 | Acc. | Mac. F1 | Acc. | Mac. F1 | Acc. | Mac. F1 | Acc. | Mac. F1 | Acc. | Mac. F1 | Acc. | Mac. F1 |
| BASE | 76.8 ±0.8* | 76.8 ±0.8* | 74.8 ±0.9* | 74.8 ±0.9* | 62.1 ±7.9* | 62.0 ±8.3* | 71.2 ±2.8* | 69.7 ±3.6* | 64.6 ±1.4* | 65.4 ±1.4* | 80.8 ±1.6* | 80.9 ±2.0* | 72.7 ±1.6* | 76.3 ±0.9* |
| MMD | 58.2 ±6.9* | 52.8 ±11.7* | 65.8 ±2.9* | 65.6 ±3.1* | 56.5 ±6.1* | 46.3 ±13.1* | 70.3 ±3.7* | 66.2 ±3.5* | 66.0 ±1.9* | 65.2 ±2.3* | 80.9 ±1.2* | 80.5 ±1.7* | 78.2 ±1.9* | 79.1 ±1.6* |
| DTS | 61.5 ±2.2* | 61.5 ±2.2* | 58.4 ±3.2* | 58.4 ±3.2* | 61.3 ±5.0* | 61.3 ±5.5* | 72.8 ±3.3* | 70.1 ±2.6* | 64.4 ±2.3* | 64.9 ±1.5* | 79.8 ±2.4* | 79.7 ±1.7* | 67.0 ±2.2* | 67.4 ±1.5* |
| IDE-VC | 53.9 ±2.2* | 53.3 ±2.7* | 67.5 ±0.9* | 67.4 ±1.0* | 58.5 ±4.5* | 58.3 ±5.0* | 73.6 ±3.1* | 73.2 ±3.4* | 65.2 ±1.3* | 65.0 ±1.7* | 80.7 ±2.0* | 80.6 ±1.4* | 74.1 ±1.8* | 76.3 ±1.1* |
| MI | 59.6 ±1.4* | 59.0 ±1.8* | 54.2 ±3.4* | 53.5 ±4.8* | 64.4 ±6.0* | 62.4 ±7.0* | 74.9 ±2.1* | 74.5 ±2.7* | 66.0 ±1.8* | 65.5 ±1.6* | 80.9 ±1.7* | 80.7 ±2.1* | 76.3 ±1.2* | 77.6 ±1.6* |
| A-CMI | 61.1 ±3.9* | 60.0 ±4.4* | 58.2 ±4.2* | 52.4 ±4.5* | 58.6 ±5.0* | 58.6 ±5.5* | 71.4 ±3.4* | 70.0 ±3.0* | 65.4 ±1.5* | 65.5 ±1.2* | 80.2 ±1.8* | 80.3 ±2.3* | 78.8 ±1.4* | 79.8 ±0.7* |
| HFS | 63.5 ±0.8* | 63.1 ±0.8* | 57.9 ±3.7* | 56.7 ±3.7* | 57.1 ±1.4* | 57.0 ±1.7* | 67.1 ±3.5* | 65.1 ±4.0* | 48.9 ±1.8* | 39.8 ±1.5* | 78.2 ±1.2* | 78.3 ±1.5* | 75.4 ±1.7* | 71.0 ±1.3* |
| SD-HC | **82.9** ±1.1 | **82.9** ±0.8 | **79.4** ±5.3 | **79.4** ±5.3 | **75.2** ±3.5 | **75.1** ±3.8 | **83.0** ±3.0 | **83.3** ±3.6 | **69.8** ±1.9 | **69.9** ±1.4 | **84.5** ±2.3 | **84.2** ±1.5 | **82.5** ±2.0 | **82.5** ±1.5 |
| **Improvement** | ↑6.1 % | ↑6.1 % | ↑4.6 % | ↑4.6 % | ↑10.8 % | ↑12.7 % | ↑8.1 % | ↑8.8 % | ↑3.8 % | ↑4.4 % | ↑3.6 % | ↑3.3 % | ↑3.7 % | ↑2.7 % |

and Canine-BG in Table 1, 2, we train under both attribute correlations and hidden correlations and report the results on test 3 (see Appendix J for full results). For other analyses, we train under $cor_h = I(m_1; a_2 | a_1) > 0$ to focus on hidden correlations. *On other datasets with unknown modes, we construct out-of-distribution (OOD) tasks under natural correlations.* By leave-one-group-out validation based on $a_2$ (user ID or operating condition), training and test sets involve non-overlapping values of $a_2$, inducing representation distribution shifts and test-time changes in hidden correlations due to confounding on disjoint $a_2$ values, e.g., different training/test users with distinct behaviors and correlations with activities. We focus on comparing the attribute prediction performance of $a_1$ under correlation shifts or distribution shifts, evaluated by accuracy (Acc.) and macro F1 score (Mac. F1), Following (Funke et al., 2022). High performance under train-test shifts can be regarded as evidence of disentanglement, as only disentangled representations can support the robust prediction of attributes under various train-test shifts. For statistical tests, each experiment is repeated using 5 varying seeds. See details in Appendix G.

**Mode Label Acquisition.** For SD-HC, we estimate mode labels with an off-the-shelf instantiation: pre-training encoder with the prediction loss of $a_1$ and running k-means on the representations $z_1$ per attribute value $\alpha$, which requires no extra loss or sub-network. See Appendix F for the detailed algorithm. The number of modes $N_m$ is shared across $\alpha$ and tuned as a hyperparameter. Extensive analysis in Appendix A demonstrates insensitivity to a range of $N_m$ choices, substantial gains from only 2% weak mode supervision, and strong cluster structure on complex time-series datasets with up to 48 modes, demonstrating effectiveness compared to other commonly adopted pre-training methods.

**Baselines and Implementations.** We compare with typical DRL methods (**MMD** (Lin et al., 2020), **DTS** (Li et al., 2022), **IDE-VC** (Yuan et al., 2021), and **MI** (Cheng et al., 2022)), and the state-of-the-art DRL methods under correlations (**A-CMI** (Funke et al., 2022) and **HFS** (Roth et al., 2023)). For reference, we include **BASE** trained with only supervised losses. See Appendix I, G, E, H for details of baselines, implementations, network architectures, and hyperparameters.

## 5.2 COMPARISON WITH BASELINE DRL METHODS

The comparison with baseline DRL methods is shown in Table 1, from which we observe:

(1) SD-HC consistently shows superiority over the compared baselines, outperforming the best baseline by an average of 5.8% and 6.1% in accuracy and macro F1 score, respectively. This indicates that unsupervised clustering can capture underlying modes on real-world data to facilitate DRL, enabling SD-HC to better disentangle representations by improving generalization ability while preserving predictive ability. Under introduced correlations, the significant advantage of SD-HC on CMNIST and Canine-BG indicates better generalization under the shifts of attribute correlations and hidden correlations. Under natural correlations, the significant advantage on UCI-HAR indicates better generalization on real-world OOD data with complex multi-modal distributions and hidden correlations.

(2) Despite considering correlations, A-CMI and HFS still fail to improve over BASE in some cases. A-CMI deals with attribute correlations, but fails under hidden correlations due to losing important mode information for attribute prediction. HFS deals with general correlations, yet it is a necessary condition for disentanglement and cannot guarantee disentanglement (Wang & Jordan, 2024).

Table 2: Comparison with variants (mean±std). The notations follow Table 1.

| Method | CMNIST | | CFashion-MNIST | | Canine-DG | | UCI-HAR | | RealWorld | | HHAR | | MFD | |
|---|---|---|---|---|---|---|---|---|---|---|---|---|---|---|
| | Acc. | Mac. F1 | Acc. | Mac. F1 | Acc. | Mac. F1 | Acc. | Mac. F1 | Acc. | Mac. F1 | Acc. | Mac. F1 | Acc. | Mac. F1 |
| SD-HC-MC | 76.7 ±1.5* | 76.7 ±1.5* | 73.6 ±1.3* | 73.6 ±1.3* | 67.2 ±3.5* | 66.9 ±3.7* | 80.2 ±3.3* | 79.7 ±4.4* | 68.4 ±0.8 | 68.0 ±0.9 | 83.8 ±1.4 | 83.2 ±1.5 | 81.1 ±2.1 | 80.2 ±2.3* |
| SD-HC-MP | 77.8 ±0.9* | 77.8 ±1.0* | 76.3 ±2.0* | 76.3 ±2.0* | 70.2 ±3.9* | 70.2 ±3.6* | 77.6 ±5.3* | 77.5 ±4.5* | 63.7 ±0.9* | 63.3 ±0.8* | 83.5 ±1.2 | 83.4 ±1.2 | 78.4 ±2.6* | 80.1 ±2.5* |
| SD-HC-A | 77.1 ±0.9* | 77.0 ±0.9* | 76.1 ±1.6* | 76.1 ±1.6* | 70.5 ±3.5* | 70.4 ±3.6* | 82.2 ±2.3 | 82.3 ±2.7 | 63.9 ±1.6* | 63.4 ±1.4* | 81.9 ±1.6* | 81.3 ±2.1* | 81.5 ±1.6 | 81.4 ±1.1 |
| SD-HC-MG | 79.7 ±1.2* | 79.7 ±1.2* | 74.3 ±1.5* | 74.3 ±1.5* | 67.9 ±3.4* | 67.6 ±3.5* | 82.2 ±2.0 | 82.8 ±2.9 | 68.4 ±1.5 | 68.8 ±2.0 | 80.6 ±1.7* | 80.2 ±2.3* | 80.3 ±1.5* | 80.4 ±1.8* |
| SD-HC-J | 76.0 ±0.9* | 75.9 ±1.2* | 76.3 ±2.5* | 76.2 ±2.2* | 70.2 ±3.5* | 69.9 ±3.6* | 79.4 ±1.6* | 79.1 ±1.6* | 66.1 ±0.8* | 65.2 ±0.8* | 80.5 ±0.7* | 80.4 ±0.8* | 79.6 ±0.9* | 80.0 ±0.5* |
| SD-HC-ID | 80.2 ±1.5* | 80.2 ±15.0* | 74.6 ±1.0* | 74.6 ±1.0* | 71.9 ±3.7* | 71.5 ±3.8* | 77.6 ±1.8* | 76.8 ±2.3* | 68.3 ±1.2 | 67.8 ±1.1 | 77.2 ±1.9* | 75.5 ±1.5* | 80.6 ±1.7* | 80.9 ±1.2 |
| SD-HC-SD | 78.3 ±1.0* | 78.3 ±1.0* | 74.9 ±1.6* | 74.9 ±1.6* | 71.8 ±3.6* | 71.7 ±3.6* | 77.4 ±1.5* | 76.8 ±1.8* | 66.2 ±1.3* | 66.6 ±1.8* | 81.0 ±2.4* | 81.2 ±1.8* | 79.2 ±1.8* | 79.2 ±1.3* |
| SD-HC | 82.9 ±1.1 | 82.9 ±0.8 | 79.4 ±5.3 | 79.4 ±5.3 | 75.2 ±3.5* | 75.1 ±3.8* | 83.0 ±3.0 | 83.3 ±3.6 | 69.8 ±1.9 | 69.9 ±1.4 | 84.5 ±2.3 | 84.2 ±1.5 | 82.5 ±2.0 | 82.5 ±1.5 |

(3) MMD, DTS, IDE-VC, and MI fail to improve over BASE in some cases, because they overlook correlations and might hurt the predictive ability of representations. Their performance degradation from BASE is especially severe on CMNIST and CFashion-MNIST under large correlation shifts.

### 5.3 COMPARISON WITH VARIANTS

We compare with the following variants: **SD-HC-MP** and **SD-HC-MC** remove the discrimination loss and mode prediction loss, respectively; **SD-HC-A** additionally minimizes attribute-based CMI for $a_2$; **SD-HC-MG** uses Marigold (Mortensen et al., 2023) instead of k-means for clustering in high-dimensional spaces; **SD-HC-J** uses iterative k-means instead of pre-trained kmeans to jointly perform clustering and disentanglement, updating mode labels every few epochs; **SD-HC-ID** and **SD-HC-SD** use individual discriminators and one shared discriminator for modes, respectively, while SD-HC shares discriminator parameters among the modes under the same attribute value. See details in Appendix E. Table 2 shows that:

(1) SD-HC-MC and SD-HC-MP consistently underperform SD-HC, showing that both discrimination loss and mode prediction loss are crucial for achieving disentanglement. This is in line with the sufficient condition for disentanglement in Proposition 2: While mode prediction loss guides the representations to preserve mode information, discrimination loss enforces conditional independence, removing redundant information about other attributes.

(2) SD-HC-A does not improve over SD-HC, probably because one independence constraint is sufficient for disentangling $a_k$, as shown in Proposition 2. Imposing additional independence constraints requires additional adversarial training steps, which might affect training stability.

(3) SD-HC-MG generally does not improve over SD-HC, indicating that k-means is effective for our representations of 128 or 512 dimensions. Marigold could be considered for representations of higher dimensions. SD-HC-J consistently underperforms SD-HC, likely due to error accumulation in clustering updates and training instability in CMI minimization from changing mode labels. This indicates that using pretrained mode labels could provide more stable mode supervision.

(4) SD-HC-ID and SD-HC-SD consistently underperform SD-HC, indicating inefficient modeling of modes. SD-HC-SD uses excessive parameter sharing across all modes, which may limit the ability to capture distinctions among modes. SD-HC-ID removes parameter sharing, which might fail to leverage the commonality among modes. In SD-HC, moderate parameter sharing is beneficial, as modes under the same attribute value share similarities while modes under different attribute values are distinct, e.g., different walking modes share similar motion patterns, which differ substantially from the patterns of standing.

### 5.4 ROBUSTNESS AGAINST NOISE AND CORRELATIONS VERSUS FULL MODE SUPERVISION

Under varying noise levels $\sigma$ and hidden correlations $cor_h$, we compare with BASE, A-CMI, and **SD-HC-T** that minimizes mode-based CMI with *ground-truth* mode labels. Figure 5 shows that:

(1) In Figure 5(a)(c), test 1, **BASE** performs well under large noise and strong hidden correlations: by over-encoding $a_2$, it compensates for noise-induced information loss, and when the hidden correlation is strong, it recovers more information. As the correlation shift enlarges from test 1 to 3, over-encoding $a_2$ turns into a disadvantage due to poor generalization. In Figure 5(c)(d), **A-CMI** performs comparably to SD-HC under $cor_h = 0$; yet its performance decreases as $cor_h$ increases, because A-CMI does not allow representations to encode shared information induced by hidden correlations, and loses more mode information as hidden correlation increases, reflecting the general behavior of DRL methods that overlook hidden correlations.

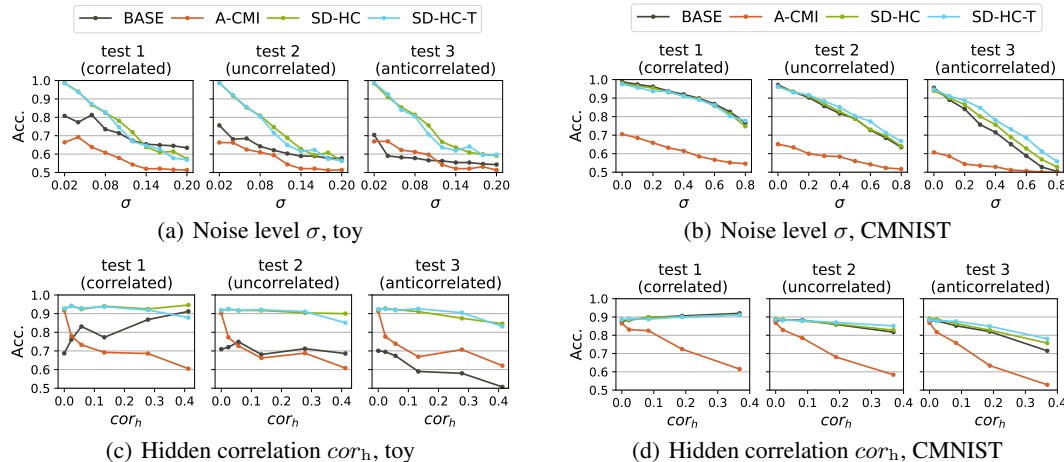

Figure 5: Comparison under varying noise level and hidden correlation on toy and CMNIST.

(2) SD-HC generally outperforms baselines, demonstrating superior robustness against noise and correlations. In Figure 5(b) test 3, SD-HC performs similarly to **SD-HC-T** at $\sigma = 0$; as noise level increases, SD-HC underperforms SD-HC-T with a stabilizing performance gap. This is due to the clustering performance decline with increasing noise (Appendix A.1). The stabilizing gap suggests that the inductive bias of unsupervised clustering leads to a stable error margin that does not widen as task difficulty increases. Notably, the performance of SD-HC-T also decreases with increasing noise, confirming that the performance drop is due to intrinsic mode ambiguity, rather than clustering errors. Meanwhile, SD-HC consistently outperforms the baselines, indicating its robust advantage.

## 5.5 METHOD INVESTIGATIONS

**Representation Distribution.** The activity representation distributions on the training set of RealWorld are visualized by t-SNE in Figure 7, which shows that: (1) **BASE** representations are separated within each activity, probably due to over-encoding user ID and learning irrelevant personalized user patterns. (2) **A-CMI** representations of different walking modes and different activities are mixed, indicating that different activities are confused due to the loss of mode information. (3) **SD-HC** representations show compactness within each activity, separation between different activities, and partition of different walking modes, indicating *Independence* from user ID and *Informativeness* of activity by encoding mode information.

**Toy Decision Boundary.** On toy data, the decision boundaries and $a_1$ prediction accuracy are shown in Figure 6, where: (1) The upper right boundaries of BASE surround the clusters at $x_2 = 1$, and its performance decreases as the correlation shift enlarges from test 1 to 3, indicating that BASE over-encodes $a_2$ and lacks generalization ability. (2) The decision boundaries of A-CMI span across the clusters at $x_2 = 0, 1$ without excluding either value, but fail to separate interleaving clusters of different modes, and its performance is low but robust across 3 test sets, indicating that A-CMI does not over-encode $a_2$, but loses important mode information. (3) The decision boundaries of SD-HC-T conform to vertical lines $x_1 = b, b \in [0, 5]$ that distinguish interleaving clusters of different modes, and SD-HC-T shows robustness and superiority across 3 test sets, indicating that SD-HC-T can learn mode information

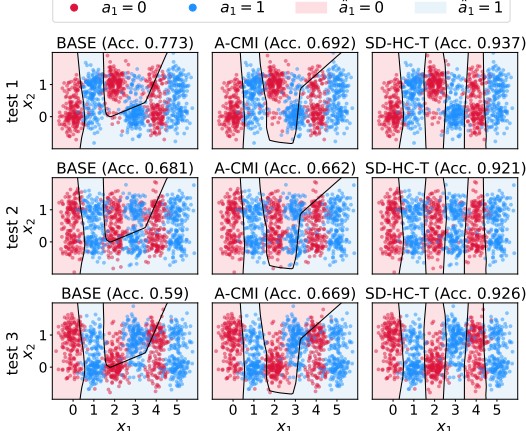

Figure 6: Toy decision boundary. Clusters centered at $x_1 = 0, 1, ...5$ are from different modes.

about $a_1$ (*Informativeness*), and exclude irrelevant information about $a_2$ (*Independence*).

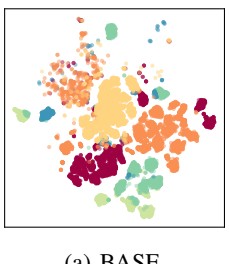 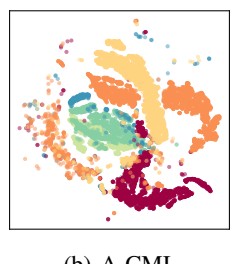 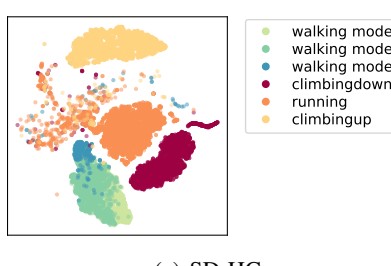

|  (a) BASE | (b) A-CMI | (c) SD-HC |

Figure 7: Activity representation distributions on RealWorld. (a), (b), and (c) show the results of four similar activities in BASE, A-CMI, and SD-HC. While activities "walking" and "climbing down" are well distinguished in SD-HC, they are confused due to losing mode information in A-CMI.

**Disentanglement Metrics.** MI and DCI-I (Eastwood & Williams, 2018) are used as disentanglement metrics under correlations, which are evaluated on the uncorrelated test set (test 2) using models trained on correlated data (Funke et al., 2022). As shown in Table 3, DRL methods generally have much lower MI than BASE, indicating *Independence*, except for HFS that allows representation correlations; SD-HC has the lowest DCI-I, indicating *Informativeness* due to encoding mode information, while DRL baselines have higher DCI-I than BASE due to hurting the predictive ability of representations.

Table 3: Disentanglement metrics. The notations follow Table 1.

| Method | MNIST | | Fashion-CMNIST | |
|---|---|---|---|---|
| | MI $\downarrow$ | DCI-I $\downarrow$ | MI $\downarrow$ | DCI-I $\downarrow$ |
| BASE | 0.548 | 0.169 | 0.645 | 0.158 |
| MMD | 0.219 | 0.386 | **0.202** | 0.210 |
| DTS | 0.328 | 0.349 | 0.339 | 0.290 |
| IDE-VC | 0.266 | 0.412 | 0.289 | 0.206 |
| MI | 0.391 | 0.372 | 0.423 | 0.377 |
| A-CMI | 0.281 | 0.332 | 0.256 | 0.353 |
| HFS | 0.482 | 0.275 | 0.491 | 0.274 |
| SD-HC | **0.212** | **0.141** | 0.261 | **0.137** |

## 6    CONCLUSIONS AND FUTURE WORK

In this paper, we propose a novel supervised DRL method under hidden correlations, SD-HC, which uses mode-based CMI minimization to achieve disentanglement for certain attributes with underlying modes and hidden correlations. Theoretically, we prove its sufficiency and show the general sufficiency of CMI minimization for disentanglement, demonstrating broad significance. Extensive experiments demonstrate the superiority of SD-HC for robust attribute prediction under varying correlation shifts, noises, and OOD tasks, confirming its practical value in real-world scenarios. Despite the advantage of SD-HC over baselines that overlook hidden correlations, we still observe a performance gap on noisy data between SD-HC and SD-HC-T with ground-truth mode labels, which is likely due to clustering errors in the pre-training stage. In future work, we plan to explore more powerful clustering approaches for discovering modes, e.g., more sophisticated pre-training strategies or joint training of clustering and disentanglement with strategies that mitigate clustering errors and preserve stability.

## 7    ETHICS STATEMENT

Our work focuses solely on scientific problems and does not involve human subjects, animals, or environmentally sensitive materials. We foresee no ethical risks or conflicts of interest.

## 8    REPRODUCIBILITY STATEMENT

We have rigorously formalized the model architecture, loss functions, and evaluation metrics through illustrations, equations, and descriptions in the main text. We provide the reproducibility details in the Appendix, including network architectures (Appendix E), training algorithm (Appendix F), dataset descriptions (Appendix G), and hyperparameters (Appendix H). We provide our source code in an anonymous link: `https://anonymous.4open.science/r/SD-HC-1FAD`, which will be publicly available upon acceptance.

## 9 USE OF LLMS

The authors use LLM solely as a general-purpose assistive tool for grammar and format refinement. LLM **does not** contribute to research ideation or experimental design. The authors take full responsibility for the content of this paper.

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

CONTENTS

# A ADDITIONAL MODEL INVESTIGATION

## A.1 CLUSTERING EVALUATION

**Clustering Metrics.** We evaluate the clustering performance with two sets of metrics. For CMNIST and CFashion-MNIST datasets with ground-truth mode labels, we use clustering accuracy (Acc.$^c$), Adjusted Rand Index (ARI), and Normalized Mutual Information (NMI) to measure the alignment between estimated mode labels and true mode labels. For other datasets with unknown modes, we use Silhouette Score (Sil.), Davies-Bouldin Index (DBI), and Calinski-Harabasz Index (CHI) to measure the intra-cluster compactness and inter-cluster separation of the cluster structure in the representation distribution. The metrics are summarized as follows:

- Clustering Accuracy (Acc.$^c$): Range $[0, 1]$; higher is better, with 1 indicating perfect alignment between estimated and true labels.

- Adjusted Rand Index (ARI): Range $[-1, 1]$ (often $[0, 1]$ in practice); higher is better, with 1 indicating perfect alignment between estimated and true labels.

- Normalized Mutual Information (NMI): Range $[0, 1]$; higher is better, with 1 indicating perfect alignment between estimated and true labels.

- Silhouette Score (Sil.): Range $[-1, 1]$; higher values mean better clustering (1 is the best), 0 indicates overlapping clusters, and negative values suggest samples are closer to another cluster than their own.

- Davies-Bouldin Index (DBI): Range $[0, \infty)$; lower values indicate better clustering with well-separated, compact clusters, with 0 being the best.

- Calinski-Harabasz Index (CHI): Range $[0, \infty)$; higher values indicate better defined and more separated clusters.

**Clustering Comparison with Different Pre-training Methods.** To investigate different pre-training methods, we compare our off-the-shelf instantiation (pre-training **BASE** with supervised attribute prediction losses only) with other commonly used pre-training methods in clustering pipelines, i.e., **AE** (autoencoder with attribute prediction losses), **InfoNCE** (BASE with attribute prediction losses and InfoNCE contrastive loss), $\beta$-**VAE**, and $\beta$-**TCVAE** (widely used variational autoencoders). The following tendencies can be observed:

(1) In general, Table 4 and 5 show that the pipeline of pre-training and k-means clustering achieves a good clustering performance, with high accuracies on image datasets and a clear indication of cluster structures on time series datasets. Pre-training methods perform differently across datasets, suggesting that the pre-training strategy could be tailored to the specific data at hand.

Table 4: Comparison of pre-training methods on CMNIST and CFashion-MNIST (mean±std). Clustering metrics are calculated by comparing to ground-truth mode labels.

| Method | CMNIST | | | CFashion-MNIST | | |
|---|---|---|---|---|---|---|
| | Acc.$^c$ ↑ | ARI ↑ | NMI ↑ | Acc.$^c$ ↑ | ARI ↑ | NMI ↑ |
| BASE | 0.758 | 0.318 | 0.286 | 0.886 | 0.611 | 0.561 |
| AE | 0.822 | 0.414 | 0.374 | 0.916 | 0.695 | 0.655 |
| InfoNCE | 0.779 | 0.332 | 0.276 | 0.855 | 0.503 | 0.460 |
| $\beta$-VAE | 0.877 | 0.568 | 0.489 | 0.648 | 0.088 | 0.208 |
| $\beta$-TCVAE | 0.852 | 0.560 | 0.482 | 0.670 | 0.115 | 0.235 |

Table 5: Comparison of pre-training methods on UCI-HAR, RealWorld, HHAR, and MFD (mean±std). Without access to ground-truth mode labels, clustering metrics are calculated by measuring intra-cluster compactness and inter-cluster separation in the representation distribution.

| Method | UCI-HAR | | | RealWorld | | | HHAR | | | MFD | | |
|---|---|---|---|---|---|---|---|---|---|---|---|---|
| | Sil. ↑ | DBI ↓ | CHI ↑ | Sil. ↑ | DBI ↓ | CHI ↑ | Sil. ↑ | DBI ↓ | CHI ↑ | Sil. ↑ | DBI ↓ | CHI ↑ |
| BASE | 0.46 | 1.03 | 1798.26 | 0.36 | 1.32 | 1521.26 | 0.33 | 1.33 | 644.44 | 0.55 | 0.69 | 4769.37 |
| AE | 0.47 | 1.02 | 1499.26 | 0.39 | 1.27 | 1644.49 | 0.24 | 1.91 | 424.15 | 0.51 | 0.79 | 3724.30 |
| InfoNCE | 0.47 | 1.04 | 1540.18 | 0.37 | 1.26 | 1490.11 | 0.33 | 1.33 | 691.70 | 0.54 | 0.73 | 3910.93 |
| $\beta$-VAE | 0.47 | 1.59 | 418.85 | 0.43 | 1.12 | 1962.74 | 0.24 | 1.82 | 422.83 | 0.54 | 0.77 | 5251.03 |
| $\beta$-TCVAE | 0.47 | 1.60 | 418.03 | 0.43 | 1.13 | 1987.38 | 0.24 | 1.88 | 430.44 | 0.55 | 0.77 | 4130.14 |
| **Total # Modes** | **48** | | | **24** | | | **12** | | | **6** | | |

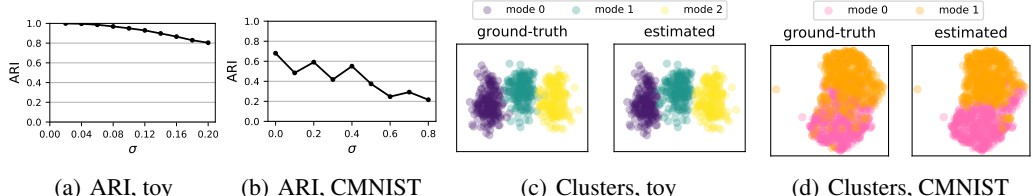

(a) ARI, toy     (b) ARI, CMNIST     (c) Clusters, toy     (d) Clusters, CMNIST

Figure 8: Clustering performance. (a) and (b) show the ARI on toy and CMNIST under varying noise levels. (c) and (d) show the true and estimated cluster assignments under $a_1 = 0$ on the raw toy data and the CMNIST representations of BASE by t-SNE (Maaten, L. V. D. and Hinton, G., 2008).

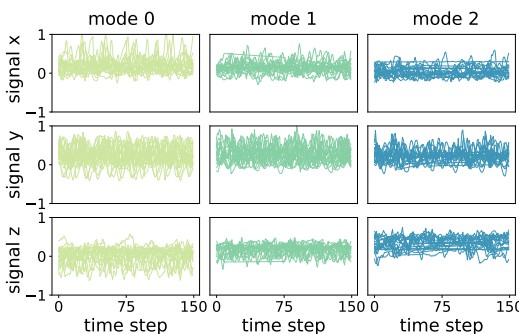

Figure 9: 3-channel accelerometer signals of three walking modes (20 random samples per mode with xyz channels). The x-axis indicates time steps, and the y-axis indicates normalized signals.

(2) Table 5 shows that time series datasets exhibit positive Silhouette Scores, low DBI values, and high CHI values, indicating the presence of underlying mode structures in each dataset. We show the total number of modes in each dataset, which is determined by hyperparameter tuning (analyzed in Appendix A.3). The valid mode structure under large numbers of modes aligns with the observation that time series attributes often display complex patterns, which may be induced by the presence of underlying modes.

**The Impact of Noise on Clustering Performance.** To complete the robustness analysis under varying noise levels in Section 5.4, we show the clustering performance under varying noise levels on toy and CMNIST datasets. The clustering quality of modes is evaluated by Adjusted Rand Index (ARI $\in [0, 1]$), with higher ARI indicating better alignment with ground truth. The results are shown in Figure 8(a)(b), with each point corresponding to the mode labels used by SD-HC at the matching setting in Figure 5(a)(b). We observe that: (1) ARI drops as the noise level increases, indicating the degradation of clustering performance with increasing noise. (2) CMNIST shows lower ARI than toy, as real data are typically more challenging for clustering. (3) The high ARI under moderate noise indicates a good clustering, showing the effectiveness of our mode discovery pipeline when the intrinsic mode structures are detectable.

**Visualizations of the Discovered Modes.** In addition, we visualize the discovered modes on the data and representation distribution of toy and CMNIST datasets in Figure 8(c)(d), where the similarity between estimated mode labels and true cluster assignments indicate a good clustering. We also visualize the data of estimated modes on the training set of RealWorld. The results in Figure 9 show that the signals of the three walking modes differ in mean values and volatility, possibly due to varying paces, strides, and postures in the walking activity. This justifies the presence of underlying modes within complex time series data.

## A.2 THE IMPACT OF MODE SUPERVISION

We control the supervision ratio $\tau$ to evaluate the impact of mode supervision: for $\tau = 0$, mode labels are obtained by unsupervised clustering; for $0 < \tau \leq 1$, $\tau \times 100\%$ true mode labels are

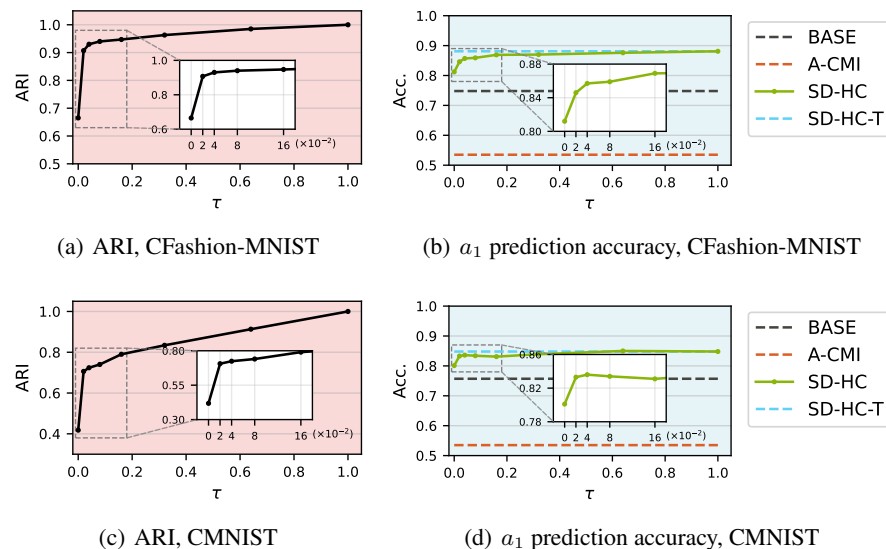

(a) ARI, CFashion-MNIST      (b) $a_1$ prediction accuracy, CFashion-MNIST

(c) ARI, CMNIST      (d) $a_1$ prediction accuracy, CMNIST

Figure 10: Comparison under varying supervision ratio on CFashion-MNIST.

provided as supervision, with the rest obtained by semi-supervised learning. The clustering quality of modes is evaluated by Adjusted Rand Index (ARI $\in [0, 1]$), with higher ARI indicating better alignment with ground truth. The ARI on the estimated mode labels and the $a_1$ prediction accuracy on test 3 (with the largest correlation shift) under varying $\tau$ are shown in Figure 10, where:

(1) Figure 10(b)(d) shows that unsupervised clustering ($\tau = 0$) provides an advantage for SD-HC over baselines, indicating that our mode discovery pipeline can discover useful mode information for disentanglement. As the amount of supervision increases ($\tau > 0$), ARI and accuracy sharply increase and soon converge, indicating that a small portion (2%) of weakly supervised labels can greatly enhance clustering and thus facilitate better disentanglement.

(2) Figure 10(b)(d) shows that, at $\tau = 0$, SD-HC exhibits a wider gap from SD-HC-T on the more complex CFashion-MNIST (6.9%) than on CMNIST (4.7%), suggesting that on complex data, estimating useful mode labels for disentanglement may be more challenging, and weak supervision may be beneficial. In practice, mode labels could be obtained via expert annotation, e.g., fine-level activity annotations for human activity data (Chan et al., 2024).

## A.3 PARAMETER SENSITIVITY

**The Number of Modes** $N_{\mathrm{m}}$. The sensitivity to the number of modes $N_{\mathrm{m}}$ under each attribute value is shown in Figure 11(a), which shows that: (1) SD-HC performs the best at the ground truth $N_{\mathrm{m}} = 2$ on CMNIST, suggesting that prior knowledge about $N_{\mathrm{m}}$ would be beneficial. (2) SD-HC performs badly at $N_{\mathrm{m}} = 1$, where mode-based CMI degrades to attribute-based CMI, causing the loss of mode information. (3) In general, SD-HC is not particularly sensitive to changes of $N_{\mathrm{m}}$ within a certain range. On CMNIST, SD-HC performs comparably under $N_{\mathrm{m}} = 2, 3, 4$, suggesting that SD-HC is robust to the changes of $N_{\mathrm{m}}$ when it is slightly larger than the ground truth ($N_{\mathrm{m}} = 2$). Probably because as long as the samples within one estimated cluster belong to the same ground-truth mode, SD-HC can preserve mode information to some extent.

In practice, hyperparameter tuning may come with high computational costs for large-scale datasets. Alternatively, we offer practical guidance to reduce the computational costs by estimating the number of modes $N_{\mathrm{m}}$ in a data-driven manner. This requires expert knowledge to choose the suitable method: For well-separated clusters, Elbow Method (Marutho et al., 2018) would be suitable for estimating $N_{\mathrm{m}}$ with k-means clustering; For complex and overlapping clusters, Bayesian Information Criterion (Watanabe, 2013) would be suitable for estimating $N_{\mathrm{m}}$ with Gaussian Mixture Models for clustering; In addition, during our pre-training stage, the number of modes can be estimated by split and merge operations with deep clustering methods (Ronen et al., 2022).

**The Weight of Mode Prediction Loss** $w_m$. The sensitivity to the weight parameter of mode prediction loss, $w_m$, is shown in Figure 11(b), which shows that: In general, SD-HC performs better

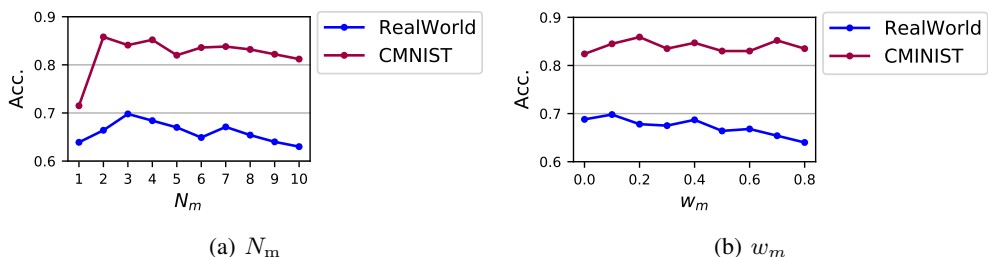

(a) $N_{\mathrm{m}}$            (b) $w_m$

Figure 11: Hyperparameter sensitivity.

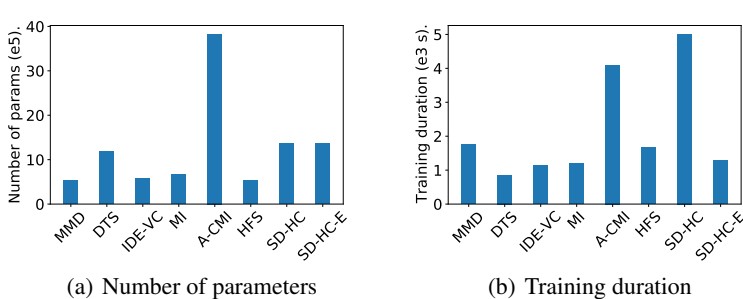

(a) Number of parameters      (b) Training duration

Figure 12: Computational complexity comparison.

at a small value of $w_m$. Theoretically, adding mode prediction loss benefits disentanglement. However, enforcing mode prediction with estimated mode labels will potentially introduce errors, as the estimated mode labels do not match the ground-truth mode labels.

## A.4 COMPUTATIONAL COMPLEXITY

Figure 12 shows the total numbers of parameters and the training durations of a single leave-one-group-out validation process (without repetition) on UCI-HAR of SD-HC and the compared methods.

In Figure 12(a), we observe that A-CMI has the most parameters, which is because A-CMI has two discriminators for minimizing conditional mutual information based on $a_1$ and $a_2$. This indicates that our method is computationally efficient w.r.t. number of parameters compared to A-CMI, which is advantageous for deployment in resource-constrained environments.

In Figure 12(b), we observe that the training durations of A-CMI and SD-HC are the longest. This training overhead would not affect real-time applications, as DRL methods are designed to generalize to unseen data, and can therefore be trained offline and deployed for real-time inference on various incoming data. However, the increased training cost can still be a concern when scaling to larger datasets.

To mitigate this issue, we design a more efficient variant, **SD-HC-E**. The long training duration of SD-HC mainly arises from the adversarial training for minimizing mode-based CMI, which involves a for-loop over modes under different attribute values and multiple discriminator update steps. For acceleration, SD-HC-E leverages vectorization to parallelize the forward computation across modes under different attribute values, and replaces the original adversarial loss with a Wasserstein GAN loss with Gradient Penalty (WGAN-GP) and Spectral Normalization (SN) (Gulrajani et al., 2017; Miyato et al., 2018) to ensure training stability with only a single discriminator update step per batch. The results in Figure 12 show that this variant substantially reduces the training duration without introducing any additional parameters. For large-scale datasets, applying this variant would be more practical and computationally efficient.

Table 6: Prediction accuracy on toy data with multiple attributes.

| **BASE** | $a_1$ | $a_2$ | $a_3$ | $a_4$ |
|---|---|---|---|---|
| test 1 | 0.965 | 0.871 | 0.999 | 0.999 |
| test 2 | 0.757 | 0.697 | 0.997 | 0.998 |
| test 3 | 0.539 | 0.585 | 0.997 | 0.994 |
| **A-CMI** | $a_1$ | $a_2$ | $a_3$ | $a_4$ |
| test 1 | 0.510 | 0.593 | 0.998 | 0.525 |
| test 2 | 0.507 | 0.537 | 0.997 | 0.507 |
| test 3 | 0.507 | 0.518 | 0.996 | 0.476 |
| **SD-HC** | $a_1$ | $a_2$ | $a_3$ | $a_4$ |
| test 1 | 0.994 | 0.797 | 0.998 | 0.999 |
| test 2 | 0.870 | 0.740 | 0.998 | 1.000 |
| test 3 | 0.752 | 0.741 | 0.997 | 0.999 |
| **SD-HC-T** | $a_1$ | $a_2$ | $a_3$ | $a_4$ |
| test 1 | 0.999 | 0.997 | 0.997 | 1.000 |
| test 2 | 0.966 | 0.982 | 0.996 | 1.000 |
| test 3 | 0.928 | 0.964 | 0.998 | 0.999 |

### A.5 EXPERIMENTS ON MULTIPLE ATTRIBUTES WITH HIDDEN CORRELATIONS

We conduct experiments on a multi-attribute toy dataset to validate the effectiveness of SD-HC in a more complex scenario.

**Data Construction.** We construct 4-dimensional toy data with 4 attributes ($a_i, 1 \leq i \leq 4$), where 2 attributes ($a_1, a_2$) exhibit underlying modes. This setting allows us to observe the impact on complex attributes with multi-modal distributions and simple attributes with uni-modal distributions. This dataset extends the simple toy dataset used in the main paper. Similarly, each data axis is controlled by one attribute, i.e., $\boldsymbol{x}_i$ is affected by $a_i$ and unaffected by other attributes $a_{-i}$. Here, $a_1$ and $a_2$ with underlying modes are constructed the same as the $a_1$ in the simple toy data, with 3 modes under each attribute value; $a_3$ and $a_4$ are constructed the same as the $a_2$ in the simple toy data. The mappings from attribute/mode labels to the corresponding data axis remain the same.

**Experiment Settings.** We use the same settings as the simple toy data, training on correlated data and evaluating on three test sets: test 1 with the same correlations, test 2 without correlations, and test 3 with anticorrelations. Complex attribute and hidden correlations are introduced in the train data, e.g., $I(a_2; a_4) = 0.07, I(a_3; a_4) = 0.13, I(m_1; a_2|a_1) = 0.14, I(m_1; a_4|a_1) = 0.36, I(m_2; a_3|a_2) = 0.28$. The task is to learn disentangled representations for each attribute. We compare SD-HC with BASE, A-CMI, and SD-HC-T. For SD-HC(-T), we use mode-based conditional mutual information (CMI) minimization for $a_1$ and $a_2$ with underlying modes, and attribute-based CMI minimization for $a_3$ and $a_4$. For A-CMI, attribute-based CMI minimization is used for all attributes.

**Results and Discussions.** The attribute prediction accuracy is reported in Table 6, showing that:

(1) **A-CMI** performs poorly on $a_4$, even though $a_4$ does not exhibit underlying modes and is easily predicted by the BASE method. This is because, under hidden correlations $I(m_1; a_4|a_1)$, minimizing attribute-based CMI for $a_1$ might degrade the representation quality for the correlated $a_4$, as indicated by Proposition 1.

(2) **SD-HC-T** outperforms SD-HC by a larger margin compared to the simple two-attribute toy data, likely because: The increased complexity with more attributes makes mode discovery harder, resulting in more errors in mode-based CMI and mode prediction losses. Reduced Informativeness in one representation can impact the disentanglement of others, as indicated by our Proposition 2. Thus, mode estimation errors for $a_1$ and $a_2$ not only harm the quality of their own representations, but also affect the representations of other attributes when they are jointly disentangled.

(3) Still, **SD-HC** generally shows **superiority** compared to BASE and A-CMI. For attributes $a_1, a_2$ with underlying modes, SD-HC explicitly accounts for hidden correlations by discovering and leveraging modes, thus better preserving mode information. For the simple attributes $a_3, a_4$, even though

they do not exhibit underlying modes, considering their hidden correlations with the modes of other attributes helps maintain their predictive accuracy in joint disentanglement.

## B PROOFS

We give the complete proof of the decomposition and propositions in the main paper using knowledge of mutual information and entropy.

Note that we use formal definitions of mutual information, where separators **semicolon ";"** and **comma ","** should be distinguished from each other. Semicolon ";" separates groups of variables whose mutual information with respect to each other is being measured, while comma "," denotes the joint distribution of the listed variables.

### B.1 PROOF OF TOTAL HIDDEN CORRELATION

**Total Hidden Correlation.** $I(m_1; a_2) = I(a_1; a_2) + I(m_1; a_2|a_1)$

*Proof.* Firstly, we prove $I(m_1; a_2) = I(m_1, a_1; a_2)$. Since each mode falls under one particular attribute value, the value of attribute is fully determined given the modes, i.e., $H(a_1|m_1) = 0$. Therefore, $H(a_1|m_1) = H(a_1|m_1, a_2) + I(a_1; a_2|m_1) = 0$, and followingly $I(a_1; a_2|m_1) = 0$, as both terms are non-negative. Hence $H(a_2|m_1) = H(a_2|m_1, a_1) + I(a_1; a_2|m_1) = H(a_2|m_1, a_1)$. Therefore, we have:

$$I(m_1, a_1; a_2) = H(a_2) - H(a_2|m_1, a_1)$$
$$= H(a_2) - H(a_2|m_1)$$
$$= I(m_1; a_2)$$

Secondly, we prove $I(m_1, a_1; a_2) = I(a_1; a_2) + I(m_1; a_2|a_1)$ by chain rule of mutual information:

$$I(m_1, a_1; a_2) = H(a_2) - H(a_2|m_1, a_1)$$
$$= H(a_2) - H(a_2|a_1) + H(a_2|a_1) - H(a_2|m_1, a_1)$$
$$= I(a_1; a_2) + I(m_1; a_2|a_1)$$

Finally, we reach $I(m_1; a_2) = I(m_1, a_1; a_2) = I(a_1; a_2) + I(m_1; a_2|a_1)$

### B.2 PROOF OF PROPOSITION 1

**Proposition 1.** *If $I(m_1; a_2|a_1) > 0$, then enforcing $I(z_1; z_2|a_1) = 0$ leads to at least one of $I(z_1; m_1) < H(m_1)$ and $I(z_2; a_2) < H(a_2)$.*

*Proof.* We prove by contradiction. Assuming $I(z_1; m_1) = H(m_1)$ and $I(a_2; z_2) = H(a_2)$ both stand, we have $H(m_1|z_1) = 0$ and $H(a_2|z_2) = 0$.

Firstly, we prove that this leads to $I(m_1; a_2; z_1; z_2|a_1) > 0$ with (1)(2)(3).

(1) Since $H(m_1|z_1) = 0$ and $H(m_1|z_1) - H(m_1|a_1, z_1) = I(m_1; a_1|z_1) \geq 0$ by definition of conditional mutual information, we have $0 \leq H(m_1|a_1, z_1) \leq H(m_1|z_1) = 0$, we have $H(m_1|a_1, z_1) = 0$. By definition, $H(m_1|a_1, z_1) = H(m_1|a_1, a_2, z_1) + I(m_1; a_2|a_1, z_1) = 0$, which gives $I(m_1; a_2|a_1, z_1) = 0$, as both terms are non-negative. Therefore:

$$I(m_1; a_2; z_1|a_1) = I(m_1; a_2|a_1) - I(m_1; a_2|a_1, z_1)$$
$$= I(m_1; a_2|a_1) > 0$$

(2) Similar to (1), since $H(a_2|z_2) = 0$ and $0 \leq H(a_2|a_1, z_2) \leq H(a_2|z_2) = 0$, we have $H(a_2|a_1, z_2) = 0$. By definition, $H(a_2|a_1, z_2) = H(a_2|m_1, a_1, z_2) + I(m1; a2|a_1, z_2) = 0$, which gives $I(m_1; a_2|a_1, z_2) = 0$, as both terms are non-negative. Therefore:

$$I(m_1; a_2; z_2|a_1) = I(m_1; a_2|a_1) - I(m_1; a_2|a_1, z_2)$$
$$= I(m_1; a_2|a_1) > 0$$

(3) Given $H(m_1|\boldsymbol{z}_1) = 0$, we have $H(m_1|\boldsymbol{z}_1) = H(m_1|\boldsymbol{z}_1, \boldsymbol{z}_2) + I(m_1; \boldsymbol{z}_2|\boldsymbol{z}_1) = 0$ and thus $H(m_1|\boldsymbol{z}_1, \boldsymbol{z}_2) = 0$, as both terms are non-negative. Similar to (1) that yields $I(m_1; a_2; \boldsymbol{z}_1|a_1) = I(m_1; a_2|a_1)$ from $H(m_1|\boldsymbol{z}_1) = 0$, we can get $I(m_1; a_2; \boldsymbol{z}_1|a_1, \boldsymbol{z}_2) = I(m_1; a_2|a_1, \boldsymbol{z}_2)$ from $H(m_1|\boldsymbol{z}_1, \boldsymbol{z}_2) = 0$ by additionally conditioning on $\boldsymbol{z}_2$. Combined with $I(m_1; a_2; \boldsymbol{z}_2|a_1) > 0$ in (2), we have:

$$I(m_1; a_2; \boldsymbol{z}_1; \boldsymbol{z}_2|a_1) = I(m_1; a_2; \boldsymbol{z}_1|a_1) - I(m_1; a_2; \boldsymbol{z}_1|a_1, \boldsymbol{z}_2)$$
$$= I(m_1; a_2|a_1) - I(m_1; a_2|a_1, \boldsymbol{z}_2)$$
$$= I(m_1; a_2; \boldsymbol{z}_2|a_1) > 0$$

Secondly, we prove $I(m_1; a_2; \boldsymbol{z}_1; \boldsymbol{z}_2|a_1) \leq 0$ with (4)(5)(6).

(4) Given $H(m_1|a_1, \boldsymbol{z}_1) = 0$ in (1), we have $H(m_1|a_1, \boldsymbol{z}_1) = H(m_1|a_1, \boldsymbol{z}_1, \boldsymbol{z}_2) + I(m_1; \boldsymbol{z}_2|a_1, \boldsymbol{z}_1) = 0$ and followingly, $I(m_1; \boldsymbol{z}_2|a_1, \boldsymbol{z}_1) = 0$, as both terms are non-negative. Therefore:

$$I(m_1; \boldsymbol{z}_1; \boldsymbol{z}_2|a_1) = I(m_1; \boldsymbol{z}_2|a_1) - I(m_1; \boldsymbol{z}_2|a_1, \boldsymbol{z}_1)$$
$$= I(m_1; \boldsymbol{z}_2|a_1) \geq 0$$

(5) Since $I(\boldsymbol{z}_1; \boldsymbol{z}_2|a_1) = 0$, we have:

$$I(m_1; \boldsymbol{z}_1; \boldsymbol{z}_2|a_1) = I(\boldsymbol{z}_1; \boldsymbol{z}_2|a_1) - I(\boldsymbol{z}_1; \boldsymbol{z}_2|m_1, a_1)$$
$$= -I(\boldsymbol{z}_1; \boldsymbol{z}_2|m_1, a_1) \leq 0$$

(6) Combine $I(m_1; \boldsymbol{z}_1; \boldsymbol{z}_2|a_1) \geq 0$ in (4) and $I(m_1; \boldsymbol{z}_1; \boldsymbol{z}_2|a_1) \leq 0$ in (5), we have $I(m_1; \boldsymbol{z}_1; \boldsymbol{z}_2|a_1) = 0$. Given $H(m_1|a_1, \boldsymbol{z}_1) = 0$ in (1) and $H(m_1|a_1, \boldsymbol{z}_1) = H(m_1|a_1, \boldsymbol{z}_1, \boldsymbol{z}_2) + I(m_1; \boldsymbol{z}_2|a_1, \boldsymbol{z}_1)$, we have $H(m_1|a_1, \boldsymbol{z}_1, \boldsymbol{z}_2) = 0$ as both terms are non-negative. Similar to (4) that yields $I(m_1; \boldsymbol{z}_1; \boldsymbol{z}_2|a_1) = I(m_1; \boldsymbol{z}_2|a_1)$ from $H(m_1|a_1, \boldsymbol{z}_1) = 0$, we can get $I(m_1; \boldsymbol{z}_1; \boldsymbol{z}_2|a_1, a_2) = I(m_1; \boldsymbol{z}_2|a_1, a_2)$ from $H(m_1|a_1, \boldsymbol{z}_1, \boldsymbol{z}_2) = 0$ by additionally conditioning on $\boldsymbol{z}_2$. Therefore:

$$I(m_1; a_2; \boldsymbol{z}_1; \boldsymbol{z}_2|a_1) = I(m_1; \boldsymbol{z}_1; \boldsymbol{z}_2|a_1) - I(m_1; \boldsymbol{z}_1; \boldsymbol{z}_2|a_1, a_2)$$
$$= -I(m_1; \boldsymbol{z}_2|a_1, a_2) \leq 0$$

This is contradictory with $I(m_1; a_2; \boldsymbol{z}_1; \boldsymbol{z}_2|a_1) > 0$. Therefore, if $I(m_1; a_2|a_1) > 0$ and $I(\boldsymbol{z}_1; \boldsymbol{z}_2|a_1) = 0$, then at least one of $I(m_1; \boldsymbol{z}_1) < H(m_1)$ and $I(a_2; \boldsymbol{z}_2) < H(a_2)$ must hold.

### B.3 PROOF OF PROPOSITION 2

**Proposition 2.** *Under the data generation process of Definition 1 ($K = 2, k = 1$), if $I(\boldsymbol{z}_1; m_1) = H(m_1)$, $I(\boldsymbol{z}_2; a_2) = H(a_2)$, and $I(\boldsymbol{z}_1; \boldsymbol{z}_2|m_1) = 0$, then $p(\boldsymbol{z}_1|\mathrm{do}(a_2)) = p(\boldsymbol{z}_1)$, i.e., $\boldsymbol{z}_1$ is the disentangled representation of $a_1$.*

Our proof for Proposition 2 is two-fold. First, from the MI terms in the proposition, a conditional independence is derived using mutual information theory in Lemma 2.1; Second, using the derived conditional independence and the assumption in the proposition, we arrive at post-interventional invariance by do-calculus in Lemma 2.2.

#### B.3.1 PROOF OF LEMMA 2.1

**Lemma 2.1.** *If $I(\boldsymbol{z}_1; m_1) = H(m_1)$, $I(\boldsymbol{z}_2; a_2) = H(a_2)$, and $I(\boldsymbol{z}_1; \boldsymbol{z}_2|m_1) = 0$, then $I(\boldsymbol{z}_1; a_2) = I(m_1; a_2)$ and $I(\boldsymbol{z}_1; a_2|m_1) = 0$.*

*Proof.* First, we prove $I(m_1; a_2) \geq I(\boldsymbol{z}_1; \boldsymbol{z}_2)$ with (1)(2).

(1) Since $H(a_2|\boldsymbol{z}_2) = 0$, we have $H(a_2|\boldsymbol{z}_2) = H(a_2|\boldsymbol{z}_1, \boldsymbol{z}_2) + I(\boldsymbol{z}_1; a_2|\boldsymbol{z}_2) = 0$, and followingly $I(\boldsymbol{z}_1; a_2|\boldsymbol{z}_2) = 0$, as both terms are non-negative. Therefore, by definition of interaction information, we have $I(\boldsymbol{z}_1; \boldsymbol{z}_2; a_2) = I(\boldsymbol{z}_1; a_2) - I(\boldsymbol{z}_1; a_2|\boldsymbol{z}_2) = I(\boldsymbol{z}_1; a_2)$. Since $I(\boldsymbol{z}_1; \boldsymbol{z}_2|m_1) = 0$, we have $I(\boldsymbol{z}_1; \boldsymbol{z}_2; a_2|m_1) = I(\boldsymbol{z}_1; \boldsymbol{z}_2|m_1) - I(\boldsymbol{z}_1; \boldsymbol{z}_2|m_1, a_2) = -I(\boldsymbol{z}_1; \boldsymbol{z}_2|m_1, a_2)$. Therefore:

$$I(\boldsymbol{z}_1; \boldsymbol{z}_2; m_1; a_2) = I(\boldsymbol{z}_1; \boldsymbol{z}_2; a_2) - I(\boldsymbol{z}_1; \boldsymbol{z}_2; a_2|m_1)$$
$$= I(\boldsymbol{z}_1; a_2) + I(\boldsymbol{z}_1; \boldsymbol{z}_2|m_1, a_2)$$
$$\geq I(\boldsymbol{z}_1; a_2)$$

(2) i. Since $H(a_2|\boldsymbol{z}_2) = 0$, we have $H(a_2|\boldsymbol{z}_2) = H(a_2|m_1, \boldsymbol{z}_2) + I(m_1; a_2|\boldsymbol{z}_2) = 0$, and followingly $I(m_1; a_2|\boldsymbol{z}_2) = 0$, as both terms are non-negative.

ii. Since $H(m_1|\boldsymbol{z}_1) = 0$, we have $H(m_1|\boldsymbol{z}_1) = H(m_1|\boldsymbol{z}_1, \boldsymbol{z}_2) + I(m_1; \boldsymbol{z}_2|\boldsymbol{z}_1) = 0$, and followingly $H(m_1|\boldsymbol{z}_1, \boldsymbol{z}_2) = 0$, as both terms are non-negative. Therefore, $H(m_1|\boldsymbol{z}_1, \boldsymbol{z}_2) = H(m_1|\boldsymbol{z}_1, \boldsymbol{z}_2, a_2) + I(m_1; a_2|\boldsymbol{z}_1, \boldsymbol{z}_2) = 0$, and followingly $I(m_1; a_2|\boldsymbol{z}_1, \boldsymbol{z}_2) = 0$, as both terms are non-negative.

iii. Given $I(m_1; a_2|\boldsymbol{z}_2) = 0$ in i. and $I(m_1; a_2|\boldsymbol{z}_1, \boldsymbol{z}_2) = 0$ in ii. as shown above, we have $I(m_1; a_2; \boldsymbol{z}_1|\boldsymbol{z}_2) = I(m_1; a_2|\boldsymbol{z}_2) - I(m_1; a_2|\boldsymbol{z}_1, \boldsymbol{z}_2) = 0$.

iv. Since $H(m_1|\boldsymbol{z}_1) = 0$, by definition of conditional mutual information, we have $H(m_1|\boldsymbol{z}_1) = H(m_1|\boldsymbol{z}_1, a_2) + I(m_1; a_2|\boldsymbol{z}_1) = 0$, and followingly $I(m_1; a_2|\boldsymbol{z}_1) = 0$, as both terms are non-negative. Thus $I(m_1; a_2; \boldsymbol{z}_1) = I(m_1; a_2) - I(m_1; a_2|\boldsymbol{z}_1) = I(m_1; a_2)$.

Given $I(m_1; a_2; \boldsymbol{z}_1) = I(m_1; a_2)$ in iv. and $I(m_1; a_2; \boldsymbol{z}_1|\boldsymbol{z}_2) = 0$ in iii., we have:

$$I(\boldsymbol{z}_1; \boldsymbol{z}_2; m_1; a_2) = I(m_1; a_2; \boldsymbol{z}_1) - I(m_1; a_2; \boldsymbol{z}_1|\boldsymbol{z}_2)$$
$$= I(m_1; a_2)$$

Given (1)(2), we have $I(m_1; a_2) = I(\boldsymbol{z}_1; \boldsymbol{z}_2; m_1; a_2) \geq I(\boldsymbol{z}_1; a_2)$

(3) We prove $I(\boldsymbol{z}_1; a_2) \geq I(m_1; a_2)$ as follows.

i. Since $H(m_1|\boldsymbol{z}_1) = 0$, we have $H(m_1|\boldsymbol{z}_1) = H(m_1|\boldsymbol{z}_1, a_2) + I(m_1; a_2|\boldsymbol{z}_1) = 0$, and followingly $I(m_1; a_2|\boldsymbol{z}_1) = 0$, as both terms are non-negative. Thus, by chain rule of mutual information, we have:

$$I(m_1, \boldsymbol{z}_1; a_2) = I(\boldsymbol{z}_1; a_2) + I(m_1; a_2|\boldsymbol{z}_1)$$
$$= I(\boldsymbol{z}_1; a_2)$$

ii. We also have:

$$I(m_1, \boldsymbol{z}_1; a_2) = I(m_1; a_2) + I(\boldsymbol{z}_1; a_2|m_1)$$
$$\geq I(m_1; a_2)$$

Given $I(m_1, \boldsymbol{z}_1; a_2) = I(\boldsymbol{z}_1; a_2)$ in i. and , $I(m_1, \boldsymbol{z}_1; a_2) \geq I(m_1; a_2)$ in ii., we have $I(\boldsymbol{z}_1; a_2) \geq I(m_1; a_2)$.

(4) Finally, given $I(m_1; a_2) \geq I(\boldsymbol{z}_1; a_2)$ with (1)(2) and $I(\boldsymbol{z}_1; a_2) \geq I(m_1; a_2)$ in (3), the equality must hold that $I(\boldsymbol{z}_1; a_2) = I(m_1; a_2)$.

Moving forward, given $I(m_1, \boldsymbol{z}_1; a_2) = I(\boldsymbol{z}_1; a_2) = I(m_1; a_2) + I(\boldsymbol{z}_1; a_2|m_1)$ in (3) and $I(\boldsymbol{z}_1; a_2) = I(m_1; a_2)$ at which we just arrived, we have $I(\boldsymbol{z}_1; a_2|m_1) = 0$.

### B.3.2  PROOF OF LEMMA 2.2

**Lemma 2.2.** *Under the data generation process of Definition 1 ($K = 2$, $k = 1$), if $I(\boldsymbol{z}_1; a_2|m_1) = 0$, then $p(\boldsymbol{z}_1|\mathrm{do}(a_2)) = p(\boldsymbol{z}_1)$.*

*Proof.* We prove this by applying Rule 3 of do-calculus based on the causal graph $G$ in Figure 3(c) (reproduced as below), which reflects the representation learning process. The rules of do-calculus are elaborated in Appendix D.2, where $\perp\!\!\!\perp$ indicates independence between variables, for arbitrary disjoint sets of nodes $X, Z, W$, $G_{\overline{X}}$ denotes the graph obtained by deleting all arrows pointing to $X$-nodes from G, and $Z(W)$ denotes the subset of $Z$-nodes that are not ancestors of any $W$-node.

Specifically, we unfold the left-hand side of $p(\boldsymbol{z}_1|\mathrm{do}(a_2)) = p(\boldsymbol{z}_1)$ and reach the right-hand side as:

$$p(\boldsymbol{z}_1|\mathrm{do}(a_2)) = \sum_{m_1} p(\boldsymbol{z}_1|\mathrm{do}(a_2), m_1)p(m_1|\mathrm{do}(a_2)) \qquad \text{(i)}$$

$$= \sum_{m_1} p(\boldsymbol{z}_1|m_1)p(m_1) \qquad \text{(ii)}$$

$$= p(\boldsymbol{z}_1) \qquad \text{(iii)}$$

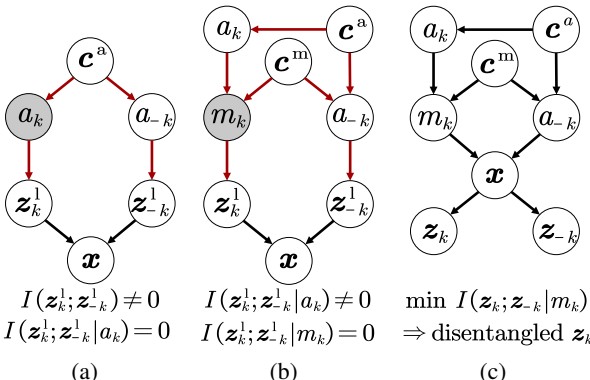

$$I(\boldsymbol{z}_k^1; \boldsymbol{z}_{-k}^1) \neq 0 \qquad I(\boldsymbol{z}_k^1; \boldsymbol{z}_{-k}^1|a_k) \neq 0 \qquad \min I(\boldsymbol{z}_k; \boldsymbol{z}_{-k}|m_k)$$
$$I(\boldsymbol{z}_k^1; \boldsymbol{z}_{-k}^1|a_k) = 0 \qquad I(\boldsymbol{z}_k^1; \boldsymbol{z}_{-k}^1|m_k) = 0 \qquad \Rightarrow \text{disentangled } \boldsymbol{z}_k$$

(a) (b) (c)

Figure 14: Causal graphs of representations under $K > 2$. (a) and (b): Data generation with *the true latent representations* $z_k^1, z_{-k}^1$, where **Red** arrows indicate the *backdoor paths* between them. (c): Representation learning that produces *the learned representations* $\boldsymbol{z}_k, \boldsymbol{z}_{-k}$.

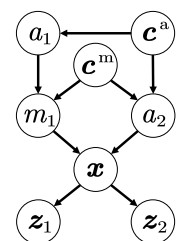

where we arrive at (i) by chain rule of probability, and then arrive at (ii) by using Rule 3 of do-calculus twice: **First**, given $I(\boldsymbol{z}_1; a_2|m_1) = 0$, we have $\boldsymbol{z}_1 \perp\!\!\!\perp a_2|m_1$ in $G$, as the mutual information between variables equals zero if and only if they are independent; for $G_{\overline{a_2(m_1)}} = G_{\overline{a_2}}$ (obtained by removing the edges pointing to $a_2$ from confounders $\boldsymbol{c}^{\mathrm{a}}, \boldsymbol{c}^{\mathrm{m}}$ in $G$), this conditional independence still holds for the following reasons (Pearl, 2009): For $\boldsymbol{z}_1$ and $a_2$, such edge removal (1) leaves the direct path $a_2 \rightarrow \boldsymbol{x} \rightarrow \boldsymbol{z}_1$ intact, not introducing any new pathway, and (2) blocks the backdoor paths $a_2 \leftarrow \boldsymbol{c}^{\mathrm{m}} \rightarrow m_1 \rightarrow \boldsymbol{x} \rightarrow \boldsymbol{z}_1$ and $a_2 \leftarrow \boldsymbol{c}^{\mathrm{a}} \rightarrow a_1 \rightarrow m_1 \rightarrow \boldsymbol{x} \rightarrow \boldsymbol{z}_1$, thus further reducing potential dependencies between $\boldsymbol{z}_1$ and $a_2$; now we satisfy the condition of Rule 3 and apply do-calculus as:

Figure 3(c) (reproduced). Causal graph of representation learning.

$$p(\boldsymbol{z}_1|\mathrm{do}(a_2), m_1) = p(\boldsymbol{z}_1|m_1) \qquad \text{Rule 3 by } \boldsymbol{z}_1 \perp\!\!\!\perp a_2|m_1 \text{ in } G_{\overline{a_2}} \text{ (representation learning)}$$

**Second**, given the causal structure in Figure 3(c) based on the data generation process of Definition 1, we satisfy the condition of Rule 3 and apply do-calculus as:

$$p(m_1|\mathrm{do}(a_2)) = p(m_1) \qquad \text{Rule 3 by } m_1 \perp\!\!\!\perp a_2 \text{ in } G_{\overline{a_2}} \text{ (elementary ingredients)}$$

Finally, we arrive at (iii) by chain rule of probability.

*Discussions.* Our proof mainly relies on two conditions: (1) there is no causal effect between $m_1$ and $a_2$, which comes from the *elementary ingredients* assumption about attributes (Suter et al., 2019), and (2) conditional independence $I(\boldsymbol{z}_1; a_2|m_1) = 0$, which is enforced upon $\boldsymbol{z}_1$ by representation learning that minimizes mode-based CMI, as proved in Proposition 2. Thereby, we conclude that *for data attributes that are elementary ingredients, disentangled representations can be learned by mode-based CMI minimization and supervised learning*.

### B.4 GENERALIZATION TO MULTIPLE ATTRIBUTES

Our theoretical results, including the necessary condition and the sufficient condition for disentanglement, can be generalized to multiple attributes. The extension mainly involves replacing $m_1, \boldsymbol{z}_1$ with $m_k, \boldsymbol{z}_k$, and replacing $a_2, \boldsymbol{z}_2$ with the joint $a_{-k}, \boldsymbol{z}_{-k}$, as the properties of mutual information and causal graphs remain the same for joint variables.

**The Necessary Condition for Disentanglement.** Figure 3(a)(b) with two attributes is extended to Figure 14(a)(b) with $K$ attributes, where the true latent representations satisfy the conditional independence as follows, yielding the necessary condition for disentanglement under hidden correlations and attribute correlations:

$$I(\boldsymbol{z}_k^1; \boldsymbol{z}_{-k}^1|m_k) = 0 \Rightarrow \textit{If } \boldsymbol{z}_k \textit{ is the disentangled representation of } a_k, \textit{ then } I(\boldsymbol{z}_k; \boldsymbol{z}_{-k}|m_k) = 0 \tag{6}$$

**The Sufficient Condition for Disentanglement.** We extend Proposition 2 to Corollary 2 for $K > 2$. The constraint $I(a_k; z_k) = H(a_k)$ is added, which is originally implied in $I(z_k; m_k) = H(m_k)$, because each mode falls under exactly one attribute value, making the attribute determined knowing the mode. In other words, all information about $a_k$ is already contained in $m_k$. In addition, the joint constraint $I(z_{-k}; a_{-k}) = H(a_{-k})$ is broken down for each $i \neq k$ into $I(z_i; a_i) = H(a_i), i \neq k$. We also extend Lemma 2.1 and 2.2 to Corollary 2.1 and 2.2 for $K > 2$, with the causal graph of representation learning depicted in Figure 14(c).

**Corollary 2.** *Under the data generation process of Definition 1, if $I(z_i; a_i) = H(a_i)$ for $i = 1, ..., K$, $I(z_k; m_k) = H(m_k)$, and $I(z_k; z_{-k}|m_k) = 0$, then $p(z_k|\mathrm{do}(a_{-k})) = p(z_k)$, i.e., $z_k$ is the disentangled representation of $a_k$.*

**Corollary 2.1.** *If $I(z_i; a_i) = H(a_i)$ for $i = 1, ..., K$, $I(z_k; m_k) = H(m_k)$, and $I(z_k; z_{-k}|m_k) = 0$, then $I(z_k; a_{-k}) = I(m_k; a_{-k})$ and $I(z_k; a_{-k}|m_k) = 0$.*

**Corollary 2.2.** *Under the data generation process of Definition 1, if $I(z_k; a_{-k}|m_k) = 0$, then $p(z_k|\mathrm{do}(a_{-k})) = p(z_k)$.*

where $-k$ indicates the set of attribute indices $\{j\}_{j \neq k}$.

Since the properties of mutual information and causal graphs remain the same for joint variables, and the MI term formulations in Corollary 2 and the causal graph structures in Figure 14 remain the same as those in Proposition 2 and Figure 3 after replacing the corresponding variables, the proofs under two attributes naturally extend to multiple attributes.

## C   DATA GENERATION PROCESS UNDER ATTRIBUTE CORRELATIONS

Following (Suter et al., 2019), the data generation process under attribute correlations is formulated in Definition 3. The causal graph of Definition 3 is depicted in Figure 15.

**Definition 3.** (Disentangled Causal Process). *Consider a causal generative model $p(x|a)$ for data $x$ with $K$ attributes $a = (a_1, a_2, ..., a_K)$ as the generative factors, where $a$ could be influenced by $L$ confounders $c = (c_1, ..., c_L)$. This causal model is called disentangled if and only if it can be described by a structural causal model (SCM) (Pearl, 2009) of the form:*

$$c \leftarrow n^{\mathrm{c}}$$
$$a_i \leftarrow h_i(S_i^{\mathrm{c}}, n_i), S_i^{\mathrm{c}} \subset \{c_1, ..., c_L\}, i = 1, ..., K \qquad (7)$$
$$x \leftarrow g(a, n^{\mathrm{x}})$$

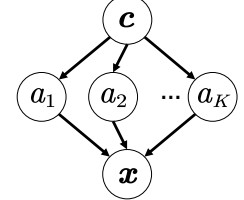

with functions $g, h_i$, jointly independent noise variables $n^{\mathrm{c}}, n^{\mathrm{x}}, n_i$, and confounder subsets $S_i^{\mathrm{c}}$ for $i = 1, ..., K$. Note that $\forall i \neq j, a_i \not\rightarrow a_j$.

Figure 15: Causal graph of data generation process with attribute correlations.

## D   CAUSALITY

### D.1   D-SEPARATION AND BACKDOOR PATHS

**Overview of Causality.**   We provide a summary of notions in causal graphs relevant to the analysis in Section 3.3, namely d-separation, blocking paths, and conditional independence. More details can be found in (Pearl, 2009).

Causal graphs are directed acyclic graphs, where nodes represent random variables and directed edges represent the causal relationships between two variables. The notion of d-separation forms the link between blocking paths in the causal graph and dependencies between random variables. A *path* in causal graphs is a sequence of consecutive edges. Consider two nodes $X$ and $Y$, $X$ and $Y$ are called *d-separated* by a set of nodes $Z$ if all undirected paths from $X$ to $Y$ are *blocked* by $Z$. Meanwhile, a path between $X$ and $Y$ is considered to be *blocked* by a set of nodes $Z$ if at least one of the following holds:

(1) The path contains a chain $X \rightarrow M \rightarrow Y$ with the mediator set $M$, and a node in $M$ is in $Z$.

(2) The path contains a fork $X \leftarrow U \rightarrow Y$ with the confounder set $U$, and a node in $U$ is in $Z$.

(3) The path contains a collider $X \rightarrow C \leftarrow Y$ with the collider node $C$, and neither $C$ or its descendant is in $Z$.

Finally, if $X$ and $Y$ are d-separated by the set $Z$, $X$ and $Y$ are conditionally independent given $Z$. A *backdoor path* between $X$ and $Y$ is the non-causal path between $X$ and $Y$ that contains at least one edge pointing at $X$ or $Y$, i.e. the path that flows backward from $X$ or $Y$. Backdoor paths introduce dependence between variables, thus they need to be blocked by controlling a node on these paths as in (1) and (2).

**Causal Graph Analysis Under Hidden Correlations.** Figure 3(b) contains three paths between $z_1$ and $z_2$. (1) The path $z_1 \rightarrow x \leftarrow z_2$ is blocked without conditioning on any variables, as long as the collider $x$ is uncontrolled. (2) The path $z_1 \leftarrow m_1 \leftarrow c^{\mathrm{m}} \rightarrow a_2 \rightarrow z_2$ is blocked if any node in the confounder set $\{m_1, c^{\mathrm{m}}, a_2\}$ is controlled. Since $c^{\mathrm{m}}$ is unobserved, controlling either $m_1$ or $a_2$ blocks this path. (3) The path $z_1 \leftarrow m_1 \leftarrow a_1 \leftarrow c^{\mathrm{a}} \rightarrow a_2 \rightarrow z_2$ is blocked if any node in the confounder set $\{m_1, a_1, c^{\mathrm{a}}, a_2\}$ is controlled. Since $c^{\mathrm{a}}$ is unobserved, controlling one of $m_1$, $a_1$, and $a_2$ blocks this path. To simultaneously block all undirected paths between $z_1$ and $z_2$, we need to control either $m_1$ or $a_2$, as controlling $a_1$ does not block path (2). That is to say, $z_1$ and $z_2$ are conditionally independent given either $m_1$ or $a_2$.

### D.2 RULES OF *do*-CALCULUS

Let $X$, $Y$, $Z$, and $W$ be arbitrary disjoint sets of nodes in a causal DAG $G$. *do*-calculus consists of three inference rules that permit us to map interventional and observational distributions to each other whenever certain conditions hold in the causal diagram $G$.

We denote by $G_{\overline{X}}$ the graph obtained by deleting from G all arrows pointing to nodes in $X$. Likewise, we denote by $G_{\underline{X}}$ the graph obtained by deleting from G all arrows emerging from nodes in $X$. To represent the deletion of both incoming and outgoing arrows, we use the notation $G_{\overline{X}\underline{Z}}$. The following three rules are valid for every interventional distribution compatible with $G$ (Pearl, 2016; 1995).

- **Rule 1**: *Insertion/deletion of observations*
$$P(y|do(x), z, w) = P(y|do(x), w), \quad \text{if } Y \perp\!\!\!\perp Z | X, W \text{ in } G_{\overline{X}}$$

- **Rule 2**: *Action/observation exchange*
$$P(y|do(x), do(z), w) = P(y|do(x), z, w), \quad \text{if } Y \perp\!\!\!\perp Z | X, W \text{ in } G_{\overline{X}\underline{Z}}$$

- **Rule 3**: *Insertion/deletion of actions*
$$P(y|do(x), do(z), w) = P(y|do(x), w), \quad \text{if } Y \perp\!\!\!\perp Z | X, W \text{ in } G_{\overline{X}\overline{Z(W)}}$$

where $\perp\!\!\!\perp$ indicates independence, and for $G_{\overline{X}\overline{Z(W)}}$, $Z(W)$ denotes the set of $Z$-nodes that are not ancestors of any $W$-node in $G_{\overline{X}}$.

## E NETWORK ARCHITECTURES

The detailed architectures of different components in SD-HC and its variants are summarized in Table 7. For independent control of each attribute, encoder $F$ uses individual subnetworks for each attribute with the same architectures. Predictors $C_i$, $C_i^{\mathrm{m}}$ share the same architectures as well. Different architectures of discriminator $D_k$ in SD-HC, SD-HC-A, SD-HC-ID, and SD-HC-SD are described separately.

## F TRAINING PROCESS

The training process of SD-HC under $K = 2$ ($a_1$ as the attribute with underlying modes) is summarized in Algorithm 1, where optimizations w.r.t. different losses are performed alternatively. The algorithm can be generalized to multiple attributes accordingly.

Table 7: Network architectures. "Discriminator($a_{in}$)" denotes discriminator with conditional input $a_{in}$. "Conv(c$i$, k$j$, s$l$)" denotes 1D convolution layer with $i$ channels, kernel size $j$, and stride $l$. "FC($i$)" denotes fully connected layer with output dimension $i$. "BN($i$)" denotes 1D batch normalization layer with feature dimension $i$. "AvgPool($i$)" denotes 1D adaptive pooling layer with output dimension $i$. "LeakyReLU($\alpha$)" denotes LeakyReLU activations with scale $\alpha$. Output dimension $d_{out}$ is set according to each prediction task. $N_1^c$ and $N_2^c$ denote the number of values for $a_1$ and $a_2$, respectively.

| Component | Method | Dataset | Architectures |
|---|---|---|---|
| Encoder subnetwork | All | Toy | FC(16) $\rightarrow$ FC(16) |
| Encoder subnetwork | All | CMNIST, CFashion-MNIST | FC(128), BN(128) $\rightarrow$ FC(128), BN(128) |
| Encoder subnetwork | All | Canine-BG | ResNet18 |
| Encoder subnetwork | All | WHAR | Conv(c128, k8, s2), BN(128) $\rightarrow$ Conv(c256, k5, s2), BN(256) $\rightarrow$ Conv(c128, k3, s1), BN(128), AvgPool(1) |
| Encoder subnetwork | All | MFD | Conv(c64, k32, s6), BN(64) $\rightarrow$ Conv(c128, k8, s2), BN(128) $\rightarrow$ Conv(c128, k8, s2), BN(128), AvgPool(1) |
| Predictor | All | All | FC($d_{out}$), Softmax |
| Discriminator($m_1$) | SD-HC | All | $N_1^c \times$ [FC(512), LeakyReLu(0.2) $\rightarrow$ FC(1), Sigmoid] for each value of $a_1$ |
| Discriminator(-) | SD-HC-A | All | $N_2^c \times$ [FC(512), LeakyReLu(0.2) $\rightarrow$ FC(1), Sigmoid] for each value of $a_2$ |
| Discriminator(-) | SD-HC-ID | All | $N_1^c \times N_m \times$ [FC(512), LeakyReLu(0.2) $\rightarrow$ FC(1), Sigmoid] for each mode under each value of $a_1$ |
| Discriminator($a_1$, $m_1$) | SD-HC-SD | All | FC(512), LeakyReLu(0.2) $\rightarrow$ FC(1), Sigmoid |

---

**Algorithm 1** The training process of SD-HC under $K = 2$

---

1: **Input:** Training set $\mathcal{D}$ with data $\boldsymbol{x}$ and attributes labels $\boldsymbol{a} = (a_1, a_2)$, the number of modes $N_m$ under each value of $a_1$, the number of epochs $E_1$ and $E_2$, and the number of steps $S_d$, $S_f$, and $S_c$
2: Initialize encoder $F^*$ and predictor $C_1^*$
3: **for** $epoch = 1$ **to** $E_1$ **do**
4:    **for** mini-batch $(\boldsymbol{x}, a_1)$ **in** $\mathcal{D}$ **do**
5:       Update $F^*$ and $C_1^*$ by minimizing $\mathcal{L}_{ac}$ in Equation 4
6:    **end for**
7: **end for**
8: Under each value of $a_1$, perform k-means clustering with the number of clusters $N_m$ on the output representations $\boldsymbol{z}_1$ of the trained encoder $F^*$, and get the estimated mode labels $m_1$
9: Initialize encoder $F$, predictors $C_1, C_2, C_1^m$, and discriminator $D_1$
10: **for** $epoch = 1$ **to** $E_2$ **do**
11:    **for** mini-batch $(\boldsymbol{x}, \boldsymbol{a})$ **in** $\mathcal{D}$ **do**
12:       **for** $step = 1$ **to** $S_c$ **do**
13:          Update encoder $F$ and predictors $C_1, C_2$ and $C_1^m$ by minimizing $\mathcal{L}_c$ in Equation 4
14:       **end for**
15:       **for** $step = 1$ **to** $S_d$ **do**
16:          Update discriminator $D_1$ by minimizing $\mathcal{L}_d$ in Equation 5
17:       **end for**
18:       **for** $step = 1$ **to** $S_f$ **do**
19:          Update encoder $F$ by maximizing $\mathcal{L}_d$ in Equation 5
20:       **end for**
21:    **end for**
22: **end for**
23: **Output:** Encoder $F$ and predictor $C_1$

---

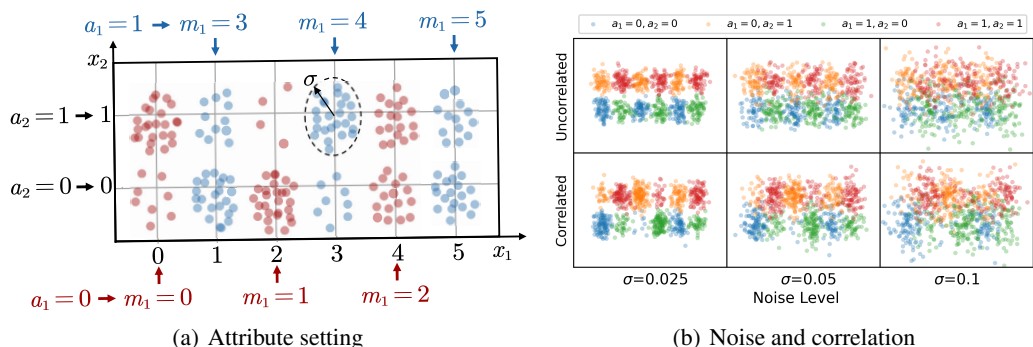

(a) Attribute setting

(b) Noise and correlation

Figure 16: Data construction of toy dataset with noise level $\sigma$.

Table 8: Conditional probability $p(a_2|m_1)$ on toy data for $cor_\mathrm{h} = 0$.

| $p(a_2\|m_1)$ | | $m_1$ | | | | | |
|---|---|---|---|---|---|---|---|
| | | 0 | 1 | 2 | 3 | 4 | 5 |
| $a_2$ | 0 | 0.5 | 0.5 | 0.5 | 0.5 | 0.5 | 0.5 |
| | 1 | 0.5 | 0.5 | 0.5 | 0.5 | 0.5 | 0.5 |

## G  DETAILS OF EXPERIMENTAL SETTINGS

### G.1  DATASETS

**Toy Dataset.** As shown in Figure 16(a), our 2-dimensional toy data have two binary attributes, with the primary attribute $a_1$ having 3 modes under each attribute value, i.e., $a_1 = 0, m_1 = 0, 1, 2$ and $a_1 = 1, m_1 = 3, 4, 5$. Data are generated through linearly mapping $m_1$ and $a_2$ to two-dimensional spaces and adding noises with noise level $\sigma$ as $\boldsymbol{x} = \boldsymbol{m}_1 \cdot [[0,0],[2,0],[4,0],[1,0],[3,0],[5,0]] + \boldsymbol{a}_2 \cdot [[0,0],[0,1]] + \boldsymbol{n}$, where vectors $\boldsymbol{m}_1$ and $\boldsymbol{a}_2$ represent the one-hot encoded values of $m_1$ and $a_2$, respectively, and $\boldsymbol{n} \sim \mathcal{N}(\boldsymbol{0}, \sigma^2\boldsymbol{I})$ represents 2-dimensional independently normally distributed noise with noise level $\sigma$. For $\boldsymbol{x} = (x_1, x_2)$, the primary attribute $a_1$ and mode $m_1$ control dimension 1, i.e., $\boldsymbol{x}_1$, and attribute $a_2$ controls dimension 2, i.e., $x_2$. An illustration of the generated data under different correlations and noise levels is given in Figure 16(b).

**CMNIST.** Colored MNIST (**CMNIST**) is constructed by coloring and occluding a subset of MNIST (Arjovsky et al., 2019), as shown in Figure 17(a). Parity check identifies whether a digit is even or odd with multiple digits under each parity value. Accordingly, attribute $a_1$ is defined as the parity of digits, i.e., $a_1 = 0, 1$ indicates "even", "odd"; attribute $a_2$ is defined as the color of digits, i.e., $a_2 = 0, 1$ indicates "red", "blue", which is often correlated with digits, e.g., a player's jersey number may be associated with a specific color in sports. $a_1$ has 2 modes under each attribute value, i.e., digits 4, 2 under parity "even" and digits 3, 9 under parity "odd". Digit noises are generated as occlusion masks with occlusion ratio as the noise level $\sigma$ (Chai et al., 2021), and coloring noises are generated as a scalar multiplier to the RGB values of the digits.

**CFashion-MNIST.** Colored Fashion-MNIST (**CFashion-MNIST**) is constructed similarly as CMNIST by coloring and occluding a subset of Fashion-MNIST (Xiao et al., 2017), as shown in Figure 17(b). This is provided as a complex counterpart of CMNIST for comparison. The correlation and noise settings are the same as CMNIST. Attribute $a_1$ is defined as the fashion styles of clothing, i.e., $a_1 = 0, 1$ indicates "sporty", "chic"; attribute $a_2$ is defined as the color of clothing, i.e., $a_2 = 0, 1$ indicates "red", "blue". $a_1$ has 2 modes under each attribute value, i.e., "sneaker" and "pullover" under style "sporty", and "sandle" and "dress" under style "chic". As a natural scenario, the color of clothing is often related to the fashion style and the specific clothing type.

Table 9: Conditional probability $p(a_2|m_1)$ on toy data for $cor_\mathrm{h} = 0.02$.

| $p(a_2\|m_1)$ | | $m_1$ | | | | | |
|---|---|---|---|---|---|---|---|
| | | 0 | 1 | 2 | 3 | 4 | 5 |
| $a_2$ | 0 | 0.6 | 0.3 | 0.6 | 0.5 | 0.6 | 0.4 |
| | 1 | 0.4 | 0.7 | 0.4 | 0.5 | 0.4 | 0.6 |

Table 10: Conditional probability $p(a_2|m_1)$ on toy data for $cor_\mathrm{h} = 0.06$.

| $p(a_2\|m_1)$ | | $m_1$ | | | | | |
|---|---|---|---|---|---|---|---|
| | | 0 | 1 | 2 | 3 | 4 | 5 |
| $a_2$ | 0 | 0.7 | 0.2 | 0.6 | 0.4 | 0.7 | 0.4 |
| | 1 | 0.3 | 0.8 | 0.4 | 0.6 | 0.3 | 0.6 |

Table 11: Conditional probability $p(a_2|m_1)$ on toy data for $cor_\mathrm{h} = 0.13$.

| $p(a_2\|m_1)$ | | $m_1$ | | | | | |
|---|---|---|---|---|---|---|---|
| | | 0 | 1 | 2 | 3 | 4 | 5 |
| $a_2$ | 0 | 0.8 | 0.1 | 0.6 | 0.3 | 0.8 | 0.4 |
| | 1 | 0.2 | 0.9 | 0.4 | 0.7 | 0.2 | 0.6 |

Table 12: Conditional probability $p(a_2|m_1)$ on toy data for $cor_\mathrm{h} = 0.28$.

| $p(a_2\|m_1)$ | | $m_1$ | | | | | |
|---|---|---|---|---|---|---|---|
| | | 0 | 1 | 2 | 3 | 4 | 5 |
| $a_2$ | 0 | 0.9 | 0 | 0.6 | 0.2 | 0.9 | 0.4 |
| | 1 | 0.1 | 1 | 0.4 | 0.8 | 0.1 | 0.6 |

Table 13: Conditional probability $p(a_2|m_1)$ on toy data for $cor_\mathrm{h} = 0.41$.

| $p(a_2\|m_1)$ | | $m_1$ | | | | | |
|---|---|---|---|---|---|---|---|
| | | 0 | 1 | 2 | 3 | 4 | 5 |
| $a_2$ | 0 | 1 | 0 | 0.5 | 0.1 | 1 | 0.4 |
| | 1 | 0 | 1 | 0.5 | 0.9 | 0 | 0.6 |

Table 14: Conditional probability $p(a_2|m_1)$ on CMNIST and CFashion-MNIST under attribute correlations and hidden correlations.

| $p(a_2\|m_1)$ | | $m_1$ | | | |
|---|---|---|---|---|---|
| | | 0 | 1 | 2 | 3 |
| $a_2$ | 0 | 0.8 | 0.05 | 0.2 | 0.95 |
| | 1 | 0.2 | 0.95 | 0.8 | 0.05 |

Table 15: Conditional probability $p(a_2|m_1)$ on CMNIST and CFashion-MNIST under only hidden correlations.

| $p(a_2\|m_1)$ | | $m_1$ | | | |
|---|---|---|---|---|---|
| | | 0 | 1 | 2 | 3 |
| $a_2$ | 0 | $corr_p$ | $1 - corr_p$ | $corr_p$ | $1 - corr_p$ |
| | 1 | $1 - corr_p$ | $corr_p$ | $1 - corr_p$ | $corr_p$ |

Table 16: Conditional probability $p(a_2|m_1)$ on Canine-BG under attribute correlations and hidden correlations.

| $p(a_2 \| m_1)$ | | $m_1$ | | | | | | | |
|---|---|---|---|---|---|---|---|---|---|
| | | 0 | 1 | 2 | 3 | 4 | 5 | 6 | 7 |
| $a_2$ | 0 | 0.1 | 0.2 | 0.3 | 0.4 | 0.9 | 0.8 | 0.7 | 0.6 |
| | 1 | 0.9 | 0.8 | 0.7 | 0.6 | 0.1 | 0.2 | 0.3 | 0.4 |

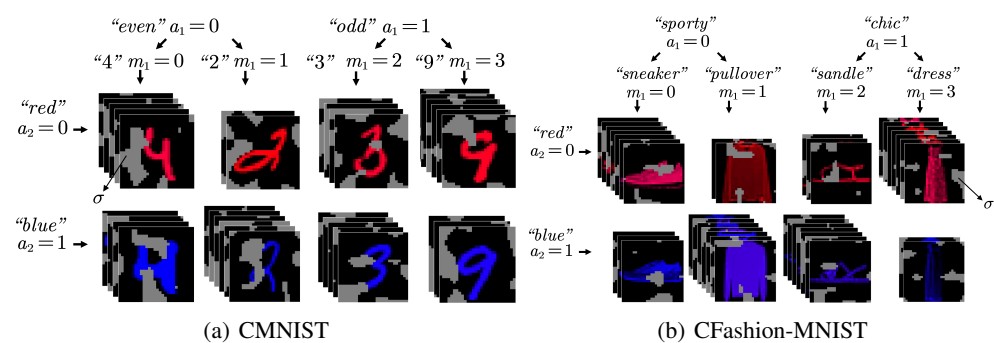

(a) CMNIST        (b) CFashion-MNIST

Figure 17: Data construction of CMNIST and CFashion-MNIST with noise level $\sigma$. The number of samples differs across $(m_1, a_2)$ combinations, exhibiting hidden correlations.

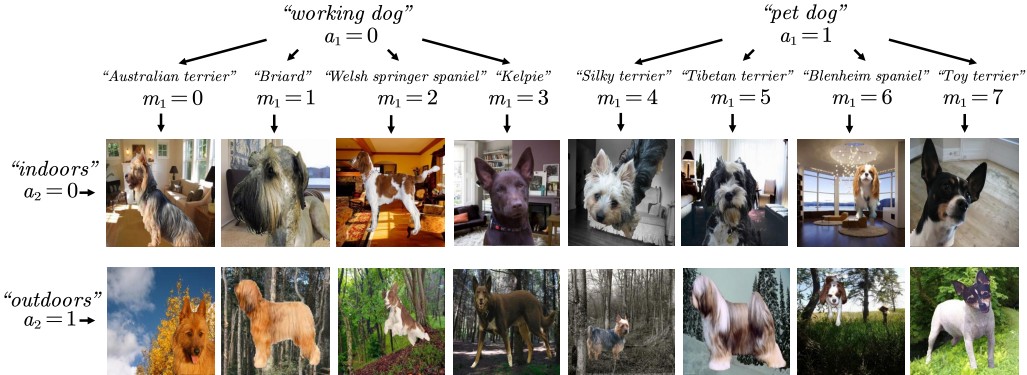

Figure 18: Data construction of Canine-BG.

**Canine-BG.** Canine-BG is constructed by combining canine images from ImageNet (Deng et al., 2009) with environmental backgrounds from the Places dataset (Zhou et al., 2018) following (Sagawa et al., 2020), as shown in Figure 18. Canines are combined with environments by the following procedure: First, SAM (Kirillov et al., 2023) is used to obtain segmentation masks of the canine image, and CLIP (Radford et al., 2021) is used to select the mask that best fits the semantics of the dog breeds with the prompt "a photo of a $\{dog\ breed\}$"; then, the canine foregrounds are combined with environment background to generate the images. Attribute $a_1$ is defined as the functional categories of canines, i.e., $a_1 = 0, 1$ indicates "working dog" and "pet dog"; attribute $a_2$ is defined as the environmental backgrounds, i.e., $a_2 = 0, 1$ indicates "indoors" ("living room" environment from Places dataset) and "outdoors" ("forest" environment from Places dataset). $a_1$ has 4 modes under each attribute value, i.e., "Australian terrier", "Briard", "Welsh springer spaniel", and "Kelpie" under category "working dog", and "Silky terrier", "Tibetan terrier", "Blenheim spaniel", and "Toy terrier" under category "pet dog". As a natural scenario, the environment of canines is often correlated to the functional categories and the specific canine breeds, e.g., working dogs are more likely to be outdoors. The correlation settings are shown in Table 16.

**Time Series Datasets.** **UCI-HAR**, **RealWorld**, and **HHAR** record wearable sensor data, from which WHAR identifies activities with variations under each activity. Accordingly, $a_1$ represents activity, $m_1$ represents unknown activity modes, and $a_2$ represents user ID, which is often correlated with activity due to personal behavior patterns. **MFD** record sensor data from bearing machines, from which machine fault diagnosis identifies machine fault types with variations under each fault type, e.g., different forms of damages. Accordingly, $a_1$ represents fault type, $m_1$ represents unknown modes of fault types, and $a_2$ represents operating conditions, which could be correlated with machine faults.

We use acceleration signals from UCI-HAR, RealWorld, and HHAR datasets and vibration signals from MFD dataset. After removing invalid values and normalizing the data by channel to be within

the range of [-1, 1], we pre-process the data by the sliding window strategy. For WHAR datasets with multiple sensors, we use the 3-axis acceleration data from the waist for UCI-HAR, the acceleration data from the chest for RealWorld, and the acceleration data from a Samsung smartphone for HHAR following (Ragab et al., 2023). Table 17 summarizes the statistics of the preprocessed data used in our experiments.

## G.2 EVALUATION PROTOCOL

On toy, CMNIST, and CFashion-MNIST datasets, correlations are introduced by sampling, and the three test sets are constructed as illustrated in Figure 19. On other datasets, we experiment under natural correlations with leave-one-group-out validation.

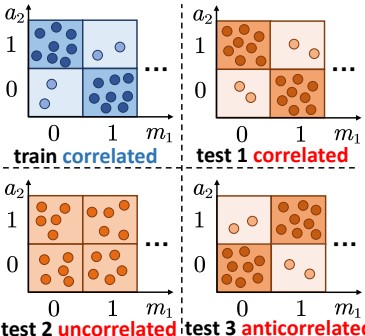

Figure 19: Train-test setup.

**Toy Dataset.** Since we focus on investigating the behavior of different methods under only hidden correlations $I(m_1; a_2|a_1) > 0$, data are set to be uniformly distributed under the values of $m_1$, $a_1$, and $a_2$, and attribute correlations do not exist, i.e., $I(a_1; a_2) = 0$. The hidden correlations are introduced by setting $p(a_2|m_1)$ to Table 8, 9, 10, 11, 12, 13 for hidden correlations $cor_h = 0, 0.02, 0.06, 0.13, 0.28, 0.41$, respectively.

**CMNIST and CFashion-MNIST.** Since we focus on investigating the behavior of different methods under various correlations, data are set to be uniformly distributed under the values of $m_1$ and $a_1$. For the comparison with baselines and variants, we introduce attribute correlations and hidden correlations by setting $p(a_2|m_1)$ to Table 14. For additional analysis, we introduce hidden correlations by setting $p(a_2|m_1)$ according to Table 15, where we set $corr_p = 0.5, 0.6, 0.7, 0.8, 0.9$ for hidden correlations $cor_h = 0, 0.02, 0.08, 0.19, 0.37$, respectively.

**Time Series Datasets.** Leave-one-group-out validation is performed, where each group is selected as the test group once, and the remaining groups serve as the training groups. Groups are obtained by dividing the data by the value of attribute $a_2$, where the number of values of $a_2$ is equal for different groups. The training and validation sets are obtained by splitting the data of the training groups by 0.8:0.2. All data of the test group form the test set. All methods are trained on the training set, tuned on the validation set, and tested on the test set.

## G.3 IMPLEMENTATION DETAILS

All methods are implemented using PyTorch (Paszke et al., 2019). We experiment with Pytorch 1.10.0+cu113 and Python 3.8.13. Model optimization is performed using Adam (Kingma & Ba, 2015). Experiments are conducted on Linux servers with Intel(R) Core(TM) i9-12900K CPUs and NVIDIA RTX 3090 GPUs.

## H HYPERPARAMETERS

The general hyperparameters are set to the following values: The number of dimensions $D$ for representations $z_i$ is set to 512 for Canine-BG and 128 for other datasets. The mini-batch size is set

Table 17: Time series dataset descriptions.

| Dataset | UCI-HAR | RealWorld | HHAR | MFD |
|---|---|---|---|---|
| $a_1$ | activity | activity | activity | incipient fault type |
| $a_2$ | user | user | user | operating condition |
| # values of $a_1$ | 6 | 8 | 6 | 3 |
| # values of $a_2$ | 30 | 15 | 9 | 4 |
| # of groups | 5 | 5 | 3 | 4 |
| # channels | 3 | 3 | 3 | 1 |
| # samples | 11711 | 36980 | 14772 | 10916 |
| window length | 128 | 150 | 128 | 5120 |
| values of $a_1$ | walking, walking upstairs, walking downstairs, sitting, standing, laying | climbing stairs up, climbing stairs down, jumping, lying, standing, sitting, running, walking | biking, sitting, standing, walking, stair up, stair down | healthy, inner-bearing damage, outer-bearing damage |

Table 18: Hyperparameter search spaces and NNI settings.

| | Item | Search space / setting |
|---|---|---|
| Hyperparameter | $w_{\mathrm{m}}$ | between [0.01, 10] |
| | $S_{\mathrm{d}}$ | [1, 3, 5, 7, 9] |
| | $N_{\mathrm{m}}$ | [2, 3, 4, 5, 6, 7, 8, 9, 10] |
| | $l_{\mathrm{c}}, l_{\mathrm{d}}, l_{\mathrm{e}}$ | [0.0001, 0.0003, 0.0005, 0.0007, 0.001] |
| NNI configuration | Max trial number per GPU | 1 |
| | Optimization algorithm | Tree-structured Parzen Estimator |

to 64 for toy data and 128 for other datasets. The number of epochs for pre-training, $E_1$, and the number of epochs for supervised DRL, $E_2$, are set to 100 and 150, respectively. The numbers of update steps $S_f$ and $S_c$ are set to 1.

Some other hyperparameters are tuned with Neural Network Intelligence (NNI)[1]. The search spaces and NNI configurations are given in Table 18. The tuned hyperparameters are set to the following values: The weight of mode prediction loss $w_{\mathrm{m}}$ is set to 0.5, 0.2, 0.5, 0.1, 0.7, 0.1, 0.01, and 0.01 on toy, CMNIST, CFashion-MNIST, Canine-BG, UCI-HAR, RealWorld, HHAR, and MFD for variants with mode prediction loss, respectively. The number of update steps $S_{\mathrm{d}}$ is set to 2, 15, 13, 9, 7, 7, 1, and 1 on toy, CMNIST, CFashion-MNIST, Canine-BG, UCI-HAR, RealWorld, HHAR, and MFD, respectively. The number of modes $N_{\mathrm{m}}$ under each value of $a_k$ is set to 3, 2, 2, 4, 8, 3, 2, and 2 on toy, CMNIST, CFashion-MNIST, Canine-BG, UCI-HAR, RealWorld, HHAR, and MFD, respectively. The initial learning rates of Adam $(l_{\mathrm{c}}, l_{\mathrm{d}}, l_{\mathrm{e}})$ are set to (0.001, 0.0007, 0.001), (0.001, 0.0003, 0.001), (0.001, 0.0003, 0.001), (0.0007, 0.0003, 0.0005), (0.001, 0.0007, 0.0005), (0.001, 0.001, 0.001), (0.001, 0.0001, 0.001), and (0.001, 0.001, 0.0005) on toy, CMNIST, CFashion-MNIST, Canine-BG, UCI-HAR, RealWorld, HHAR, and MFD, respectively.

# I  BASELINES

We focus on comparing different independence constraints, and leave out the other components in the original baseline implementations, e.g., different architectures. For fair comparisons, all methods share the same encoder structure and train with alternative update steps, which is the same as SD-HC. The baselines are summarized below:

- **MMD** (Lin et al., 2020) minimizes the Maximum Mean Discrepancy between different distributions in the subspace of one attribute under different values of another attribute.

- **DTS** (Li et al., 2022) adversarially trains attribute predictors to make one attribute unpredictable from the representations of another.

---

[1] https://github.com/microsoft/nni

Table 19: Full comparison with baselines on CMNIST dataset (mean±std, in percentage). The notations follow Table 1.

| Method | Test 1 (correlated) | | Test 2 (uncorrelated) | | Test 3 (anticorrelated) | |
| --- | --- | --- | --- | --- | --- | --- |
| | Acc. | Mac. F1 | Acc. | Mac. F1 | Acc. | Mac. F1 |
| BASE | **90.1** ±0.3 | **90.1** ±0.3 | 83.1 ±0.3* | 83.1 ±0.3* | 76.8 ±0.8* | 76.8 ±0.8* |
| MMD | 64.0 ±11.4* | 58.9 ±16.0* | 61.4 ±8.8* | 56.2 ±13.3* | 58.2 ±6.9* | 52.8 ±11.7* |
| DTS | 69.9 ±4.1* | 69.9 ±4.1* | 65.1 ±2.9* | 65.1 ±2.9* | 61.5 ±2.2* | 61.5 ±2.2* |
| IDE-VC | 63.2 ±3.1* | 62.9 ±3.1* | 58.8 ±2.5* | 58.5 ±2.8* | 53.9 ±2.2* | 53.3 ±2.7* |
| MI | 66.4 ±1.8* | 66.0 ±1.8* | 62.8 ±1.9* | 62.4 ±1.9* | 59.6 ±1.4* | 59.0 ±1.8* |
| A-CMI | 72.2 ±7.2* | 71.2 ±8.1* | 66.8 ±4.9* | 65.8 ±5.8* | 61.1 ±3.9* | 60.0 ±4.4* |
| HFS | 81.1 ±1.4* | 80.9 ±1.4* | 72.5 ±1.2* | 72.3 ±1.2* | 63.5 ±0.8* | 63.1 ±0.8* |
| SD-HC (ours) | 88.6 ±0.5 | 88.6 ±0.8 | **85.9** ±0.9 | **85.9** ±1.0 | **82.9** ±1.1 | **82.9** ±0.8 |
| **Improvement** | ↓ 1.5 % | ↓ 1.5 % | ↑2.8 % | ↑2.8 % | ↑6.1 % | ↑6.1 % |

Table 20: Full comparison with baselines on CFashion-MNIST dataset (mean±std, in percentage). The notations follow Table 1.

| Method | Test 1 (correlated) | | Test 2 (uncorrelated) | | Test 3 (anticorrelated) | |
| --- | --- | --- | --- | --- | --- | --- |
| | Acc. | Mac. F1 | Acc. | Mac. F1 | Acc. | Mac. F1 |
| BASE | **93.6** ±0.3 | **93.6** ±0.3 | 84.2 ±0.8 | 84.2 ±0.8 | 74.8 ±0.9* | 74.8 ±0.9* |
| MMD | 91.7 ±0.3 | 91.7 ±0.3 | 79.0 ±1.3* | 79.0 ±1.3* | 65.8 ±2.9* | 65.6 ±3.1* |
| DTS | 83.6 ±7.7* | 83.6 ±7.7* | 71.0 ±5.0* | 71.0 ±5.0* | 58.4 ±3.2* | 58.4 ±3.2* |
| IDE-VC | 90.0 ±1.2 | 90.0 ±1.2 | 79.4 ±1.0* | 79.4 ±1.0* | 67.5 ±0.9* | 67.4 ±1.0* |
| MI | 70.6 ±9.8* | 70.2 ±11.3* | 62.3 ±5.5* | 61.8 ±7.5* | 54.2 ±3.4* | 53.5 ±4.8* |
| A-CMI | 72.0 ±11.3* | 70.0 ±12.8* | 64.7 ±6.1* | 60.8 ±8.1* | 58.2 ±4.2* | 52.4 ±4.5* |
| HFS | 86.2 ±3.4* | 86.1 ±3.5* | 72.6 ±1.7* | 72.1 ±2.1* | 57.9 ±3.7* | 56.7 ±3.7* |
| SD-HC | 93.3 ±5.1* | 93.3 ±5.1* | **86.3** ±4.6* | **86.3** ±4.6* | **79.4** ±5.3* | **79.4** ±5.3* |
| **Improvement** | ↓ 0.3 % | ↓ 0.3 % | ↑ 2.1 % | ↑ 2.1 % | ↑ 4.6 % | ↑ 4.6 % |

Table 21: Full comparison with baselines on Canine-BG dataset (mean±std, in percentage). The notations follow Table 1.

| Method | Test 1 (correlated) | | Test 2 (uncorrelated) | | Test 3 (anticorrelated) | |
| --- | --- | --- | --- | --- | --- | --- |
| | Acc. | Mac. F1 | Acc. | Mac. F1 | Acc. | Mac. F1 |
| BASE | **85.6** ±5.3 | **85.6** ±6.3 | 72.0 ±5.5* | 72.0 ±6.1* | 62.1 ±7.9* | 62.0 ±8.3* |
| MMD | 56.4 ±3.7* | 45.6 ±11.0* | 56.7 ±4.9* | 45.6 ±11.9* | 56.5 ±6.1* | 46.3 ±13.1* |
| DTS | 77.9 ±4.0* | 77.9 ±4.5* | 71.4 ±3.5* | 71.4 ±3.8* | 61.3 ±5.0* | 61.3 ±5.5* |
| IDE-VC | 80.2 ±3.0* | 80.2 ±3.5* | 70.2 ±3.2* | 70.0 ±3.6* | 58.5 ±4.5* | 58.3 ±5.0* |
| MI | 69.0 ±6.0* | 67.7 ±7.0* | 67.7 ±5.5* | 65.9 ±6.5* | 64.4 ±6.0* | 62.4 ±7.0* |
| A-CMI | 76.8 ±4.5* | 76.8 ±5.0* | 68.1 ±4.0* | 68.1 ±4.5* | 58.6 ±5.0* | 58.6 ±5.5* |
| Hausdorff | 82.0 ±2.8* | 81.9 ±2.9* | 69.1 ±1.8* | 69.0 ±2.0* | 57.1 ±1.4* | 57.0 ±1.7* |
| SD-HC | 84.8 ±3.0 | 84.8 ±3.2 | **80.6** ±2.8 | **80.5** ±3.0 | **75.2** ±3.5 | **75.1** ±3.8 |
| **Improvement** | ↓ 0.8 % | ↓ 0.8 % | ↑ 8.6 % | ↑ 8.5 % | ↑ 10.8 % | ↑ 12.7 % |

- **IDE-VC** (Yuan et al., 2021) minimizes the unconditional MI between the representations of different attributes by adversarially training a predictor that predicts the representations of one attribute from those of another.

- **MI** (Cheng et al., 2022) and **A-CMI** (Funke et al., 2022) minimize the unconditional mutual information and the attribute-based conditional mutual information between the representations of different attributes, respectively. These two methods minimize MI by adversarially training an unconditional or conditional discriminator as the proposed method. We train two discriminators for A-CMI to minimize conditional mutual information based on both $a_1$ and $a_2$ as in (Funke et al., 2022).

- **HFS** (Roth et al., 2023) minimizes the Hausdorff distance between two representation sets to factorize the supports of different representation subspaces, where we use Euclidean distance as the distance measure between different representations from the same subspace.

## J   FULL RESULTS ON CMNIST AND CFASHION-MNIST DATASET

The full comparisons with baselines on CMNIST, CFashion-MNIST, and Canine-BG datasets are presented in Table 19, Table 20, and Table 21, respectively, from which we observe that the advantage of SD-HC increases as correlation shift increases from test 1 to test 3. Detailed method behaviour of BASE, A-CMI, and SD-HC are analyzed in Section 5.4.

