# OpenReview forum: "Supervised Disentanglement Under Hidden Correlations"
_ICLR.cc/2026/Conference — Submitted to ICLR 2026_

### Official Review · Reviewer_nRdb · 2025-10-29

**Soundness:** 2
**Presentation:** 3
**Contribution:** 2
**Rating:** 4
**Confidence:** 3

**Summary:**

The paper introduces a novel supervised disentangled representation learning method, SD-HC, designed to handle correlations between the sub-populations under one attribute and other attributes. The authors propose minimizing a mode-based conditional mutual information to achieve disentanglement while preserving crucial mode-level information that existing methods lose. They provide a theoretical proof that their approach is a sufficient condition for disentanglement under a specified causal data generation process and demonstrate its superior generalization performance through extensive experiments.

**Strengths:**

The paper identifies and formalizes the novel and important problem of hidden correlations in disentanglement, with the motivation being very clear (Fig. 1). The main theoretical contribution, proving the sufficiency of mode-based CMI for disentanglement under these conditions, is a significant step beyond prior work that only established necessity. The empirical evaluation is extensive and rigorous, spanning multiple datasets, OOD generalization tasks, and thorough ablations. The results convincingly demonstrate the superiority of the proposed method in scenarios with complex data distributions.

**Weaknesses:**

The method’s success relies heavily on an initial unsupervised clustering step (k-means on pre-trained features), making it a two-stage rather than end-to-end approach. Because of this, its overall performance depends on how well that clustering work, something that can easily fail if the data isn’t clearly separable or the pre-training isn’t strong.

It also introduces an additional hyperparameter: the number of modes per attribute. The paper shows that getting this wrong significantly hurts performance, meaning users need to tune it carefully. While the authors offer some estimation strategies, it still adds extra complexity and room for error.

Another issue is conceptual. The paper’s 'modes' could just as easily be seen as sub-categories or hierarchical labels. The authors don’t justify why their approach is better than existing hierarchical or fine-grained learning methods.

From a practical standpoint, the method may be computationally heavy: it requires pre-training, clustering for each attribute, and an adversarial training loop, making it hard to scale.

Finally, the theoretical grounding assumes that attributes are independent causal factors, an unrealistic simplification in many real-world settings, although common in disentangled feature learning.

**Questions:**

- The method's reliance on a two-stage process seems to be its main practical weakness. Have you considered methods for end-to-end learning that could jointly discover the modes (e.g., via deep clustering objectives) and optimize the disentanglement objective simultaneously? This could potentially make the method more robust and elegant.
- How does the performance compare to a baseline that treats modes as distinct sub-classes in a standard classification task without any explicit disentanglement objective? This would help clarify whether the performance gain comes from discovering the modes, the specific CMI objective, or both.
- In the causal graph, the key insight is that conditioning on $m_1$ blocks the backdoor path through $c^m$. What is the intuition for why $c^m$ cannot directly influence $x$? Is it plausible that the same unobserved confounder that affects the walking mode also directly affects the raw sensor readings (x) in a way not mediated by the mode $m_1$?
- The discriminator parameter sharing strategy across modes proved highly effective. Could you provide more intuition for this? Is it primarily a regularization effect to combat data sparsity within each mode, or does it enforce a shared structure on what it means to be independent of other attributes, regardless of the specific mode?

---

> ### Author Response · Authors · 2025-11-22
> **Response to Reviewer nRdb (Part 1)**
>
> We sincerely thank the reviewer for the thoughtful and detailed feedback, as well as the recognition of the novelty of our problem formulation, the strength of our theoretical contribution, and the thoroughness of our empirical evaluation. We have carefully addressed the concerns in the response below and highlighted the corresponding improvements made in our paper.
>
> ### **W1,2 & Q1: About the two-stage process and robustness to clustering**
>
> We appreciate the reviewer’s thoughtful remarks regarding our two-stage approach and its practical implications. We appreciate the opportunity to clarify the practical considerations of our method that are linked to our theoretical contributions.
>
> Firstly, we wish to emphasize that the core contribution of our work is theoretical:
>
> * We establish the necessary conditions and **the first sufficient conditions** of disentanglement under hidden correlations, which generalize to multiple attributes, and various correlation types and correlation strengths.
> * Our results apply to data with attribute correlations, thus serving as a general guarantee for CMI minimization as the sufficient independence constraint for disentanglement, demonstrating a broader significance.
> * We show that only one conditional independence constraint is needed per attribute, substantially reducing computational overhead when only a subset of attributes is relevant to downstream tasks.
>
> Our framework is mainly designed to **rigorously implement the sufficient conditions** based on mode labels. While we acknowledge that imperfect clustering may create a performance gap compared to full ground-truth supervision, our method maintains a strong advantage over baselines that *overlook hidden correlations*. Below, we address the reviewer's concerns in detail:
>
> 1. ***Framework design and practical considerations:***
> Our goal of methodology design is to provide a model-agnostic framework that **strictly implements the sufficient conditions based on mode labels**. We perform pre-training on **attribute prediction loss** alone and use k-means clustering to obtain mode labels. This process requires no extra loss or sub-network, and is computationally affordable. Although this process shows success across extensive experiments, we acknowledge that it depends on the quality of mode labels and might require domain expertise for selecting more suitable mode estimation methods for extremely challenging tasks. Nonetheless, as pre‑training and clustering techniques are mature and widely studied across various research areas, adapting them to particular tasks is **well supported by existing literature**.
>
> 2. ***Insensitivity to the number of modes within a certain range:***
> The model is affected by the number of modes $N_m$, but remains robust within a reasonable range of this hyperparameter. For example, when $N_m = 2, 3, 4, 5$ on RealWorld dataset (**$\underline{\text{Figure 11(a), Appendix A.3, page 19}}$**), the performance is stable, allowing some tolerance for estimation errors. In practice, the estimation strategies, as the reviewer noted, may provide effective guidance for selecting suitable values, limiting the tuning overload to a manageable level.
>
> 3. **Advantage over baselines on complex image data (additional):**
> In response to the concern regarding data that are not "clearly separable", we add experiments on a dataset, **Canine-BG**, constructed from the complex ImageNet dataset. This new dataset involves canine functional categories as attribute $a_1$, canine breeds as modes $m_1$ under each value of $a_1$, and environmental background as attribute $a_2$. Correlations are introduced to reflect the natural relationships between different dogs and their environments, e.g., different breeds have different probabilities of being seen indoors or outdoors.
>
>     * On this new dataset, SD-HC **outperforms the best baseline by 10.8\% in accuracy and 12.7\% in macro F1 score**, confirming its advantage in complex settings without clear mode structures. The revisions to include dataset constructions and additional experiments are summarised at the end of the response to **W1,2 & Q1**.
>
>     * In addition, we'd like to mention that the datasets used in our original submission also include complex patterns, e.g., the complex image data on CFashion-MNIST with **various clothing items**, as well as complex time series data with **temporal dynamics and periodic patterns**, as visualized in **$\underline{\text{Figure 9, Appenix A.1, page 18}}$**.

---

> ### Author Response · Authors · 2025-11-22
> **Response to Reviewer nRdb (Part 2)**
>
> 4. ***Robustness on challenging data:***
> In addition, we'd like to further highlight one key finding regarding data that aren't **clearly separable**, as mentioned in the review. As shown in **$\underline{\text{Figure 5 (a)(b), Section 5.4, page 9}}$**, when noise levels increase and cause the modes to be inseparable, we observe that:
>     * The performances of **all methods degrade**, including SD-HC and the baselines. This reflects an *intrinsic noise-induced difficulty common to all methods* rather than a specific weakness of clustering.
>     * Notably, SD‑HC generally **outperforms baselines that overlook hidden correlations** across varying noise levels, demonstrating its robust generalization and resilience to perturbations arising from noisy data.
>     * Also, the performance gap between **SD‑HC** (unsupervised clustering) and **SD‑HC‑T** (full ground‑truth supervision) **stablizes under large noises**. This suggests that the inductive bias introduced by unsupervised clustering leads to a *consistent, bounded error margin* instead of amplifying under greater task difficulty.
>
>     **The main reason for our advantage:** Our method is designed for data with intrinsic, meaningful modes, which ensures the discovery of mode structures to a certain extent, giving advantage to SD-HC that explicitly models modes over baselines that overlook them. While noises inevitably obscure these modes, such negative impact affects all methods comparably, thus does not diminish the advantage of SD-HC.
>
> 5. ***Weak supervision as a promising remedy:***
> In addition, we analyze the effect of **weak mode supervision** in **$\underline{\text{Figure 10, Appendix A.2, page 19}}$**, finding that **a small portion (2%)** of ground-truth mode labels can **significantly boost performance**, approaching the level of full supervision. This suggests that, in practical scenarios, acquiring a small number of reliable mode labels can serve as weak‑supervision to maintain strong performance without incurring substantial annotation costs.
>
> 6. ***End-to-end learning as a meaningful future research direction:***
> We appreciate the insightful suggestion regarding joint training with deep clustering objectives. We implement an additional variant **SD-HC-J** that jointly discovers modes via iterative k-means during training, using the original mode prediction loss as a deep clustering loss.
>
>     The results below show performance degradation of this variant, likely due to error accumulation in clustering updates and training instability in CMI minimization from changing mode assignments. These results suggest that effective end-to-end learning may require **specialized optimization techniques and stability-preserving strategies**, an interesting direction for future work but beyond the scope of our current study. In this work, we focuse on providing a **theoretically grounded framework that implements the sufficient conditions of disentanglement** for handling hidden correlations.
>
>
> | Dataset & Metric | CMNIST Acc. | CMNIST Mac. F1 | CFashion-MNIST Acc. | CFashion-MNIST Mac. F1 | UCI-HAR Acc. | UCI-HAR Mac. F1 | RealWorld Acc. | RealWorld Mac. F1 | HHAR Acc. | HHAR Mac. F1 | MFD Acc. | MFD Mac. F1 |
> |-|-|-|-|-|-|-|-|-|-|-|-|-|
> | SD-HC-J  | 0.760 ±0.9 | 0.759 ±1.2 | 0.763 ±2.5 | 0.762 ±2.2 | 79.4 ±1.6 | 79.1 ±1.6 | 66.1 ±0.8 | 65.2 ±0.8 | 80.5 ±0.7 | 80.4 ±0.8   | 79.6 ±0.9 | 80.0 ±0.5  |
> | SD-HC  | 82.9 ±1.1 | 82.9 ±0.8 | 79.4 ±5.3 | 79.4 ±5.3 | 83.0 ±3.0 | 83.3 ±3.6 | 69.8 ±1.9 | 69.9 ±1.4 | 84.5 ±2.3 | 84.2 ±1.5 | 82.5 ±2.0 | 82.5 ±1.5 |
>
> In response to this comment, we have revised **$\underline{\text{Section 1, Introduction, page 2}}$** to better articulate our core theoretical contributions and practical considerations, **$\underline{\text{Section 5.1, 5.2, 5.3}}$** and **$\underline{\text{Appendix E, G, J}}$** to include the additional variant and additional experiments and descriptions on the new dataset, and **$\underline{\text{Section 6, Conclusions and Future Work, page 10}}$** to include our future research directions. We believe these revisions strengthen the paper and hope they fully address the concern.

---

> ### Author Response · Authors · 2025-11-22
> **Response to Reviewer nRdb (Part 3)**
>
> ### **W3 & Q2: Comparison to hierarchical methods**
>
> We appreciate this insightful point. We agree that the modes can be viewed as sub-categories. However, existing hierarchical disentanglement methods are based on the assumption that attributes are independent and leverage independence constraints for disentanglement [1,2], without considering attribute correlations or hidden correlations. As analyzed in **$\underline{\text{“Disentanglement Under Attribute Correlations”, Section 2, Related Work, page 2}}$**, disentanglement methods that rely on the independence constraints fail under correlations, which makes these hierarchical disentanglement methods **fail under hidden correlations** w.r.t. the modes.
>
> SD-HC has been compared with the following variants that ablate the CMI objective and the mode prediction (classification) objective:
> * **SD-HC-MP** removes the discrimination loss for CMI minimization (keeping the mode prediction loss).
> * **SD-HC-MC** removes the mode prediction loss (keeping the discrimination loss).
>
> Please refer to **$\underline{\text{Table 2, Section 5.3, page 8}}$** for the results. Both variants show performance degradation compared to SD-HC, suggesting that **both objectives are essential** for achieving disentanglement, which aligns with our Proposition 2 (the Sufficient Condition for Disentanglement). Mode prediction loss makes the model explicitly encode the underlying modes, while discrimination loss removes redundant information about other attributes for disentanglement.
>
> In response to this comment, we have expanded analyses in our paper to more thoroughly discuss the role of these losses in **$\underline{\text{Section 5.3, page 8}}$**.
>
> [1] Benchmarks, algorithms, and metrics for hierarchical disentanglement, ICML 2021.
>
> [2] Hierarchical generative modeling for controllable speech synthesis, ICLR 2019.
>
> ### **W4: Computational complexity**
> We thank the reviewer for this comment. The training duration is mainly increased by the adversarial training for minimizing mode-based conditional mutual information (CMI). Since we only pre-train with attribute prediction losses, the pre-training process does not take up a long time. The computational efficiency of adversarial training can be improved by leveraging **parallelization and acceleration techniques**. This is implemented as an additional and more efficient variant **SD-HC-E**, with the changes elaborated below:
>
>    * Originally, we use a FOR loop on the modes under different attribute values to minimize mode-based CMI, which takes up more time. For speeding up this process on large datasets, **parallelizing using vectorization** can greatly reduce training duration without affecting the forward computation, effectively reducing from **K forward computations** (where K is the number of attribute values) to **a single forward computation per batch**.
>    * Originally, we update discriminator for several steps (the number of steps ranges from 1 to 15) in the adversarial training. To eliminate this overload, we substitute the adversarial loss with Wasserstein GAN loss with Gradient Penalty (WGAN-GP) and Spectral Normalization (SN) [3,4]. This combination provides strong Lipschitz constraints, enabling stable training with **a single discriminator update step per batch**, while also **accelerating convergence**.
>
> | Method | SD-HC (Original) | SD-HC-E (Accelerated) |
> |-|-|-|
> |Training Time| 5012.24 s | 1274.81 s |
>
> The results above show that this variant **shortens the training time by approximately 3.93 times** on UCI-HAR dataset, yielding **a comparable duration with more efficient DRL methods**, which have been incorporated in **$\underline{\text{Appendix A.4, page 20}}$**. Codes for this variant will be made available on our GitHub repository upon acceptance.
>
> [3] Improved training of Wasserstein GANs, NIPS 2017.
>
> [4] Spectral normalization for generative adversarial networks, ICLR 2018.

---

> ### Author Response · Authors · 2025-11-22
> **Response to Reviewer nRdb (Part 4)**
>
> ### **W5 & Q3: Causal structures and assumptions**
>
> We thank the reviewer for the comment regarding the causal structures and assumptions. We address the concerns point-by-point as follows.
>
> 1) **Assumption: Attributes with no causal effects between them:** We would like to clarify that **this is not equivalent to the independence assumption**. We assume each attribute is an elementary ingredient that **has no causal effect on other attributes**, meaning they can be altered independently through external interventions, while still **allowing for correlations**.
>
>     * As insightfully noted in the review, this is the standard assumption in disentangled representation learning (DRL). Although some real-world cases involve causally dependent attributes (e.g., bad weather leading to increased traffic congestion), our assumption remains valid for a wide range of **supervised disentanglement tasks**.
>
>     * For example, in human activity recognition, the **activity type** (class) and **user identity** (domain) may appear correlated, as some users perform certain activities more often. Nevertheless, **a user’s identity does not causally determine which activities they perform**, nor does engaging in specific activities determine user identity, indicating a relationship with no direct causal effects.
>
> 2) **Causal effect of confounders:**
> We appreciate the reviewer’s comment regarding the potential effect of confounders on the data. We follow the common formulations for DRL under correlations, where the variables that directly generate the data are regarded as **generative factors or attributes** of the data; Confounders, on the other hand, are typically introduced to **capture or explain correlations among these generative factors**, as adopted in prior works [3,4,5].
>
>     For example, personal habits can induce correlations between users and activities, i.e., which user tends to perform certain activities more often. However, the observed sensory signal windows only capture the co-occurrence of attribute combinations (i.e., the user and the current activity), rather than directly revealing the underlying causal mechanisms.
>
> We hope this clarification helps to explain our causal reasoning.
>
> [3] Robustly disentangled causal mechanisms: Validating deep representations for interventional robustness, ICML 2019.
>
> [4] Desiderata for representation learning: A causal perspective, JMLR 2024.
>
> [5] Disentanglement and generalization under correlation shifts, CoLLAs 2022.
>
> ### **Q4: Insights on discriminator parameter sharing**
>
> We thank the reviewer for the question regarding discriminator architectures, which is the key to efficiently minimizing mode-based CMI. First, we clarify the details of these variants:
>
> * To investigate the **discriminator architecture** of **SD‑HC**, we design two variants, **SD‑HC‑ID** and **SD‑HC‑SD**, that use *individual discriminators* and *one shared discriminator* for all modes, respectively, while **SD‑HC** uses one discriminator for each attribute value, *sharing discriminator parameters among the modes under the same attribute value* (see **$\underline{\text{Appendix E, page 27}}$** for detailed architecture).
>
> The experimental results show that:
>
> * **SD‑HC** consistently outperforms both variants, indicating a good balance in discriminator parameter sharing. In contrast, excessive parameter sharing across all modes may limit the discriminator’s ability to capture differences among modes, whereas no parameter sharing might fail to exploit the commonality shared by the modes under the same attribute value; both might lead to inefficient modeling.
> * For example, in human activities, different walking modes share similar motion patterns, where moderate parameter sharing could be benificial, whereas walking and other activities (e.g., standing) are significantly different, and sharing parameters between distinct modes under these activities might make it hard to learn specific discrimination tasks for each mode.
>
> We have expanded the discussions in **$\underline{\text{Section 5.3, page 8}}$** for a more thorough analysis. We hope this response adequately addresses the questions raised.
>
> ---
>
> We sincerely appreciate the reviewer's constructive comments and believe that the additional experiments and analyses further demonstrate the practicality and scalability of our method. We believe the revisions made to the manuscript significantly strengthen the work, and we hope the reviewer finds our responses and revisions satisfactory and supports our work with a more favorable assessment.

---

### Official Review · Reviewer_XBAQ · 2025-11-01

**Soundness:** 2
**Presentation:** 3
**Contribution:** 2
**Rating:** 4
**Confidence:** 4

**Summary:**

The paper presents SD‑HC, a supervised disentangled representation learning method designed for settings where attributes are correlated and, where multi‑modal structure (modes) exists under a given attribute. It first estimates mode labels for each attribute value, then minimizes a mode‑conditioned conditional mutual information so that representation subspaces disentangle while preserving both attribute and mode information. The authors provide formal results that this mode‑based CMI suffices for disentanglement under their causal data‑generation assumptions, and report experiments on toy data and real‑world datasets indicating improved robustness to distribution and train-test correlation shifts compared with prior DRL baselines.

**Strengths:**

- The paper is well written and easy to follow.
- The motivation is clear and important. Addressing hidden correlations in multimodal attributes is essential to make DRL practical.
- The paper includes theoretical analysis and the empirical method itself is intuitive and simple.

**Weaknesses:**

- The main results are reported with F1 and Accuracy, which do not directly assess disentanglement without any justifications. These metrics can be high even when representations are entangled. Although the authors also report MI and DCI-I at the end of Section 5, DCI-D, which is a direct measure for disentanglement and a more common disentanglement metric in the literature, is not provided.

- While the advantage of using supervision (i.e., GT labels) in disentanglement representation learning is that it can guarantee identifiability of GT factors under mild assumptions [1] (which could not be guaranteed in unsupervised setting as shown in [2]), this paper formalizes disentanglement via statistical independence (Definition 2) and does not establish identifiability of the underlying factors.

**Reference**

[1] Locatello  et al., Weakly-Supervised Disentanglement Without Compromises, in ICML 2020.

[2] Locatello  et al., Challenging common assumptions in the unsupervised learning of disentangled representations, in ICML 2019.

**Questions:**

- In L183-185, it is correct that $I(z_2; a_2) < H(a_2)$ implies that $z_2$ loses attribute information for predicting $a_2$ but does $I(z_1; m_1) < H(m_1)$ imply $z_1$ loses mode information for predicting $a_1$ (not $m_1$)?
- When the GT labels for mode information are not given (which is common), we should decide an arbitrary number of clusters, which could be differ from the GT number of modes. Are those theoretical guarantees still hold for wrong number of clusters?
- In Table 18, under Test 1 where train–test correlations are identical, the baseline shows the highest accuracy, outperforming the proposed method. What explains this gap? While it is expected that the baseline benefits when the correlation structure matches training, the proposed method’s inferior performance appears to indicate a loss of predictive information, which seems to contradicts the paper’s claim that predictive information is preserved. Moreover, in realistic deployments with careful dataset curation, correlation shift may be minimal or absent; in such cases, the baseline could be preferable.

---

> ### Author Response · Authors · 2025-11-22
> **Response to Reviewer XBAQ (Part 1)**
>
> We sincerely thank the reviewer for affirming our motivation and theoretical contributions. We greatly appreciate the time and effort invested in reviewing our work and providing detailed and constructive feedback, which we found very valuable and helpful. We hope the following responses clarify the raised concerns.
>
> ### **W1 & Q1: About evaluation metrics**
>
> We thank the reviewer for the comment regarding evaluation metrics. We agree that this is an important aspect. Specifically, accuracy and macro‑F1 are used to evaluate the robustness of attribute prediction under various train-test distribution shifts, serving as metrics for assessing disentanglement [2], as elaborated below:
>
> 1) **Our goal: Supervised disentanglement towards robust attribute prediction:** We would like to emphasize that our supervised disentanglement approach is primarily designed to improve the robustness of attribute prediction under distribution shifts. This aligns with **common applications of supervised disentanglement**, where the goal is to learn representations that robustly predict attribute values on unseen test data, e.g., training on known users and using the model to predict the activity of unseen users in activity recognition [1]. Accuracy and macro‑F1 are widely adopted to measure whether the model can robustly predict attribute values in such challenging cases.
>
> 2) **Disentanglement reflected by accuracy metrics under train-test shifts:** We fully agree that high accuracy and macro-F1 alone do not guarantee disentanglement when train and test distributions are **similar**. However, under the **significant distribution shifts and correlation shifts** as described in **$\underline{\text{“Evaluation Protocols”, Section 5.1, page 6}}$**, only **disentangled** representations can maintain a high performance. In contrast, entangled representations tend to overfit to training data and fail to generalize. Thus, following [2], we use robust prediction performance as a practical and meaningful indicator of disentanglement under train-test shifts.
>
> 3) **MI as a more suitable metric for supervised disentanglement than DCI-D:** While DCI-D is well-suited for evaluating **unsupervised disentanglement**, where each dimension does not correspond to a specific factor, it is less appropriate in our **supervised disentanglement setting**, where each representation vector is explicitly learned for a specific attribute via label supervision. Here, a more direct measure of disentanglement is the mutual information (MI) between the representations of different attributes, which captures the degree of dependency between them [2]. In addition, most disentanglemet metrics are based on the assumption that attributes are independent, thus **cannot directly apply** to correlated data [2,3]. Following [2], we train the model on correlated data, and calculate disentanglement metrics (MI and DCI-I) on an uncorrelated test set, as described in **$\underline{\text{“Disentanglement Metrics”, Section 5.5, page 10}}$**.
>
> We have incorporated the reasons for using accuracy and macro-F1 in **$\underline{\text{“Evaluation Protocols”, Section 5.1, page 6}}$** to more clearly articulate the rationale behind our choice of metrics. We hope this clarification helps.
>
> [1] Latent independent excitation for generalizable sensor-based cross-person activity recognition, AAAI 2021.
>
> [2] Disentanglement and generalization under correlation shifts, CoLLAs 2022.
>
> [3] Challenging common assumptions in the unsupervised learning of disentangled representations, ICML 2019.
>
> ### **W2: Identifiability**
> We thank the reviewer for raising the comment regarding identifiability. We fully agree that supervised disentanglement can provide identifiability guarantees under certain assumptions, as pointed out in the review.
>
> While identifiability is a valuable theoretical property, it is not the focus of our study. Following the well‑established definitions of disentanglement in [4,5], **our goal is to investigate what constraints can lead to disentanglement in such an interventional sense**, and what learning objectives can strictly enforce these constraints in a supervised setting. To the best of our knowledge, we are **the first to derive sufficient conditions for disentanglement under this formalization**, providing a theoretical foundation for how disentangled representations can be attained through appropriately designed objectives. We believe this offers a meaningful step toward practical and theoretically grounded supervised disentanglement, even as identifiability remains an open and active research direction.
>
> [4] Robustly disentangled causal mechanisms: Validating deep representations for interventional robustness, ICML 2019.
>
> [5] Desiderata for representation learning: A causal perspective, JMLR 2024.

---

> ### Author Response · Authors · 2025-11-22
> **Response to Reviewer XBAQ (Part 2)**
>
> ### **Q1: On the meaning of $I(z_1;m_1)<H(m_1)$**
>
> We thank the reviewer for this question and for the scientific rigor. The observation that “$I(\mathbf{z}_1; m_1) < H(m_1)$ indicates $\mathbf{z}_1$ loses mode information for predicting $m_1$” is accurate. Our original statement stems from the following insight about the relationship between mode and attribute prediction:
>
> As discussed in **$\underline{\text{line 61, Section 1, Introduction, page 2}}$**, the mode information ($m_1$) can be crucial for accurately predicting the attribute ($a_1$), as modes under different attribute values may form adjacent or interleaved clusters. In such cases, successful attribute prediction requires capturing these local mode structures. Therefore, the loss of mode information might further lead to degraded performance in predicting $a_1$. Relevant empirical analyses are in **$\underline{\text{Figure 6,7, Section 5.5, page 9}}$**.
>
> In response to the comment, we have revised the statement in **$\underline{\text{“A-CMI Fails Under Hidden Correlations”, Section 3.3, page 4}}$** to improve clarity and precision. We hope the updated explanation addresses the concern.
>
> ### **Q2: About the number of clusters**
> We thank the reviewer for raising this practical consideration regarding the number of clusters. Our theoretical guarantees are derived based on the ground-truth modes. In practice, an incorrectly specified number of clusters may indeed affect the degree of disentanglement achieved, but we find the error to be manageable within a certain range in the sensitivity analysis.
>
> The sensitivity analysis on this hyperparameter is in **$\underline{\text{“The Number of Modes", Appendix A.3, page 19}}$**. We find that the model exhibits **robust performance within a reasonable range of mode numbers**, and noticeable degradation only occurs when $N_m$ deviates substantially from the ground-truth value. This suggests a degree of insensitivity to moderate estimation errors. In addition, we also discuss **estimation methods** for the number of modes, which may help guide the selection and ease the practical burden of choosing this hyperparameter.
>
> ### **Q3: High BASE performance on the correlated test 1**
>
> We thank the reviewer for this insightful question and for the careful examination of our results. This model behaviour has been discussed in **$\underline{\text{point (1), Section 5.4, page 8}}$**, as elaborated below:
>
> Specifically, **BASE** refers to the baseline trained only with supervised attribute prediction losses, and without disentanglement constraints. Under hidden correlations, BASE tends to *over‑encode* $a_2$ in representation $z_1$. This can be interpreted as overfitting to the hidden correlations between $m_1$ and $a_2$ in the training data. Consequently, it achieves higher accuracy on **Test 1**, where train-test correlations are identical, because leveraging information about $a_2$ compensates for the noise‑induced information loss about $m_1$. However, this reliance on the training correlations fails upon correlation shifts, causing **severe performance degradation** on **Test 2 (uncorrelated)** and **Test 3 (anticorrelated)**, i.e., this advantage of BASE is **not robust**.
>
> For instance, on CMNIST, there may exist hidden correlations where digit 3 has an 80% probability of being blue, and digit 4 has an 80% probability of being red (assuming the number of samples from these digits is comparable). When the BASE model cannot confidently recognize whether a digit is 4 or 3, it might exploit the digit color to make a better guess. For example, **predicting “4” when seeing a red digit**. This strategy works well when the test data follow the same correlations as the training data, but fails when the correlations change, e.g., when digit 4 becomes 80% blue and digit 3 is 80% red. In such cases, relying on color leads to systematic prediction errors, explaining the **limitation of entangled representations**.
>
> In response to the question, we have further clarified this in **$\underline{\text{Appendix J, page 35}}$** by adding notes to direct readers to the corresponding discussion in the main paper. We hope this clarification helps.
>
> ---
>
> We sincerely thank the reviewer for their thorough assessment and thoughtful comments, which have helped us clarify several important points in the manuscript. We are grateful for the opportunity to enhance the clarity, and believe the paper has benefited from this feedback. We hope the reviewer finds our responses and revisions satisfactory and supports our work with a more favorable assessment.

---

### Official Review · Reviewer_uBsu · 2025-11-01

**Soundness:** 3
**Presentation:** 2
**Contribution:** 2
**Rating:** 6
**Confidence:** 3

**Summary:**

The paper addresses the challenge of supervised disentangled representation learning under hidden correlations in multi-modal data. Existing DRL methods often assume attribute independence, or at most handle observed attribute correlations via conditional mutual information. However, these methods can lose important mode information in the presence of hidden correlations. The authors propose SD-HC, which introduces mode-based conditional mutual information minimization to preserve both attribute and mode information. The paper provides theoretical guarantees (necessary and sufficient conditions for disentanglement under hidden correlations), introduces an architecture-agnostic framework, and validates the approach on toy and six real-world datasets. Results show consistent improvements over baselines in both accuracy and generalization under correlation shifts.

**Strengths:**

The theoretical contributions are significant, establishing both necessary and sufficient conditions for disentanglement under attribute and hidden correlations. This is the first formal proof showing sufficiency of conditional mutual information (CMI) minimization in this context, and the work generalizes to multiple attributes, varying correlation strengths, and both uni-modal and multi-modal distributions. The causal formulation and mode-based approach are rigorous, showing a deep understanding of the interplay between attribute and mode information in disentanglement. The proposed SD-HC method demonstrates strong performance across diverse datasets, including toy data, image datasets (CMNIST, CFashion-MNIST), and real-world wearable human activity recognition datasets. The method consistently outperforms state-of-the-art baselines, including A-CMI, HFS, and other DRL approaches, particularly under distribution shifts and correlation changes. The ablation studies convincingly show the importance of the mode-based CMI and discriminator components, highlighting that the design aligns with the theoretical guarantees. The extensive experiments, visualization of learned representations, and robustness analysis under noise and correlation shifts add further credibility to the claims.

**Weaknesses:**

While SD-HC provides strong theoretical and empirical contributions, the reliance on pre-estimated mode labels introduces an additional external step that may affect reproducibility and performance, particularly for datasets where the mode structure is unclear or highly complex. Although the paper shows some insensitivity to the number of modes, the selection of hyperparameters for mode estimation can still influence results, and the method’s dependence on clustering quality may limit applicability in high-dimensional or highly noisy settings.

Another limitation is the scalability and computational complexity of SD-HC. The framework requires adversarial training with mode-based CMI minimization, discriminators, and mode predictors, which may increase training time compared to simpler DRL methods. While the method is claimed to be architecture-agnostic, the experiments focus mainly on moderate-dimensional representations and small to medium-sized datasets; it remains unclear how SD-HC performs on very large-scale datasets or in real-time applications where computational resources are constrained.

**Questions:**

Can the SD-HC framework be efficiently scaled to very large datasets or real-time applications, and what strategies could mitigate the increased computational complexity from adversarial training and mode-based CMI minimization?

---

> ### Author Response · Authors · 2025-11-22
> **Response to Reviewer uBsu (Part 1)**
>
> We sincerely thank the reviewer for the thoughtful and affirmative evaluation of our work. We are deeply gratified by the reviewer's recognition of our theoretical contributions, the rigor of our causal formulation, and the strong performance of our approach. We are truly encouraged by this positive assessment and have prepared the following clarifications to address the remaining points.
>
> ---
>
> ### **W1: About the reliance on pre‑estimated mode labels and applicability to complex data**
>
> We thank the reviewer for the constructive comment about mode estimation, and appreciate the opportunity to clarify our methodological choices. Firstly, we'd like to highlight that our core contribution is theoretical.
>
> **Theoretical focus as our core contribution:**
> As the reviewer insightfully noted, our core contribution lies in establishing the theoretical foundations for disentanglement under hidden correlations. In addition to the reviewer's in-depth summary of our theoretical strengths, we wish to supplement two theoretical advances:
>
> 1. Our results generalize to data under only attribute correlations, serving as a general guarantee for CMI minimization as the sufficient independence constraint for disentanglement, and demonstrating a broader significance. (See **$\underline{\text{“Scope of Applicability: Key Assumptions and Generalizations”, Section 3.3, page 5}}$**)
> 2. Our results show that only one conditional independence constraint is needed for disentangling one attribute (based on its mode or attribute values), saving computational overload for supervised disentanglement when only a subset of attributes is relevant to the downstream tasks. (See **$\underline{\text{“Theoretical Contributions”, Section 3.3, page 5}}$**)
>
> **Addressing Mode Estimation Concerns:**
> Our framework is mainly designed to rigorously implement the sufficient conditions *based on mode labels*. While we acknowledge that mode estimation errors create a gap from full ground-truth mode supervision, presenting opportunities for improvement, our advantage remains strong over baselines that *overlook hidden correlations*. We appreciate the reviewer's suggestion to consider this aspect more carefully, and provide clarifications about robustness analyses on mode estimation, promising remedies, and additional experiments as follows.
>
> 1. **Advantage over baselines on complex image and time series data (original):**
> In our original submission, the experiments demonstrate SD-HC's advantage over baselines that overlook hidden correlations on complex image data on CFashion-MNIST with **various clothing items**, as well as complex time series data. **Time series** exhibit highly **complex temporal dynamics and periodic patterns**, as visualized in **$\underline{\text{Figure 9, Appenix A.1, page 18}}$**. The total number of modes on time series dataset also illustrates substantial complexity, as summarized below (from **$\underline{\text{Table 5, Appendix A.1, page 17}}$**).
>
> | Dataset | UCI‑HAR | RealWorld | HHAR | MFD |
> |:-|:-:|:-:|:-:|:-:|
> | Attribute $a_1$ | human activity | human activity | human activity | machine faults |
> | # Modes      | 48 | 24 | 12 | 6 |
>
> 2. **Advantage over baselines on complex image data (additional):**
> In response to the concern regarding **high-dimensional** data, we've added experiments on a dataset, **Canine-BG**, constructed from the complex ImageNet dataset, where the representation dimension is 518. This new dataset involves canine functional categories as attribute $a_1$, canine breeds as modes $m_1$ under each value of $a_1$, and environmental background as attribute $a_2$. Correlations are introduced to reflect the natural relationships between different dogs and their environments, e.g., different breeds have different probabilities of being seen indoors or outdoors.
>
>     On this new dataset, SD-HC **outperforms the best baseline by 10.8\% in accuracy and 12.7\% in macro F1 score**, confirming its advantage in high-dimensional, complex settings. Corresponding revisions are summarized at the end of the response to **W1**.

---

> ### Author Response · Authors · 2025-11-22
> **Response to Reviewer uBsu (Part 2)**
>
> 3. **Stable gap from full ground-truth supervision and advantage over baselines under noises:**
> In response to the concern regarding "unclear" mode structure and "highly noisy settings", we'd like to highlight the key findings from **$\underline{\text{Figure 5 (a)(b), Section 5.4, page 9}}$**. We observe that, when noise levels increase and cause the modes to be unclear:
>    * The performance of **all methods degrade**, including SD-HC and the baselines. This reflects an *intrinsic noise-induced difficulty common to all methods* rather than a specific weakness of clustering.
>    * Notably, SD‑HC generally **outperforms baselines that overlook hidden correlations** across varying noise levels, demonstrating its robust generalization and resilience to perturbations arising from noisy data.
>    * Also, the performance gap between **SD‑HC** (unsupervised clustering) and **SD‑HC‑T** (full ground‑truth supervision) **stablizes under large noises**. This suggests that the inductive bias introduced by unsupervised clustering leads to a *consistent, bounded error margin* instead of amplifying under greater task difficulty.
>
>    **The main reason for our advantage:** Our method is designed for data with intrinsic, meaningful modes, which ensures the discovery of mode structures to a certain extent, giving advantage to SD-HC that explicitly models modes over baselines that overlook them. While noises inevitably obscure these modes, such negative impact affects all methods comparably, thus do not diminish the advantage of SD-HC.
>
> 4. **Weak supervision as a promising remedy:**
> In addition, we analyze the effect of weak mode supervision in **$\underline{\text{Figure 10, Appendix A.2, page 19}}$**, finding that a **small portion (2%)** of ground-truth mode labels can **significantly boost performance**, approaching the level of full supervision. This suggests that, in practical scenarios, acquiring a small number of reliable mode labels can serve as weak‑supervision to maintain strong performance without incurring substantial annotation costs.
>
> 5. **Future directions:** Finally, we acknowledge that clustering quality is pivotal and view the pursuit of improved strategies as a promising future direction. We are committed to actively exploring more advanced techniques to further close the performance gap compared to ground-truth mode labels, thereby enhancing the practical utility of our framework.
>
> In response to this comment, we have revised the **$\underline{\text{Section 1, Introduction, page 2}}$** to better articulate our core theoretical contributions, **$\underline{\text{Section 5.1, 5.2, 5.3}}$** and **$\underline{\text{Appendix E, G, J}}$** to include additional experiments and descriptions on the new dataset, and **$\underline{\text{Section 6, Conclusions and Future Work, page 10}}$** to include our future research directions. We believe these revisions strengthen the paper and hope they fully address the concern.

---

> ### Author Response · Authors · 2025-11-22
> **Response to Reviewer uBsu (Part 3)**
>
> ### **W2&Q1: About computational complexity**
>
> We thank the reviewer for this comment regarding practical ability. SD‑HC is **well suited for real‑time applications**, as only inference is required during deployment, and SD‑HC can be effectively scaled to large datasets by leveraging **parallelization and acceleration techniques** for adversarial training, as elaborated below:
>
> 1) **Real-time applications:** Disentangled representation learning (DRL) methods are designed to obtain representations that generalize to data with distribution or correlation shifts. Therefore, DRL models can be **trained offline and directly applied to various incoming data in real applications**. This is computationally efficient, as only the forward inference is performed in real time. This setting aligns with our experiments, where the test data exhibit correlation shifts or form out‑of‑distribution tasks with the training data.
>
> 2) **Large datasets:** As the reviewer insightfully noted, the training duration is mainly increased by the adversarial training for minimizing mode-based conditional mutual information (CMI). We have implemented an accelerated variant **SD-HC-E** for larger datasets with the following changes:
>
>    * Originally, we used a FOR loop on the modes under different attribute values to minimize mode-based CMI, which takes up more time. For speeding up this process on large datasets, **parallelizing using vectorization** can greatly reduce training duration without affecting the computations, effectively reducing from **K forward computations** (where K is the number of attribute values) to **a single forward computation per batch**.
>    * Originally, we update discriminator for several steps (the number of steps ranges from 1 to 15) in the adversarial training. To eliminate this overload, we substitute the adversarial loss with Wasserstein GAN loss with Gradient Penalty (WGAN-GP) and Spectral Normalization (SN) [1,2]. This combination provides strong Lipschitz constraints, enabling stable training with **a single discriminator update step per batch**, while also **accelerating convergence**.
>
> | Method | SD-HC (Original) | SD-HC-E (Accelerated) |
> |-|-|-|
> |Training Time| 5012.24 s | 1274.81 s |
>
> As shown above, this variant **shortened the training time by approximately 3.93 times** on UCI-HAR dataset, yielding a **comparable duration with more efficient DRL methods**, which has been incorporated in **$\underline{\text{Figure 12, Appendix A.4, page 20}}$**. Codes for this variant will be made available on our GitHub repository upon acceptance.
>
> [1] Improved training of Wasserstein GANs, NIPS 2017.
>
> [2] Spectral normalization for generative adversarial networks, ICLR 2018.
>
> ---
>
> We sincerely appreciate the reviewer's constructive comments and believe that the additional experiments and analyses further demonstrate the practicality and scalability of our method. We believe the revisions made to the manuscript significantly strengthen the work.

---

### Official Review · Reviewer_bvj9 · 2025-11-06

**Soundness:** 2
**Presentation:** 2
**Contribution:** 3
**Rating:** 4
**Confidence:** 3

**Summary:**

The paper first identifies the limitations of previous supervised DRL approaches, which fail to consider the mode information of attributes for prediction. It then provides a theoretical analysis that establishes a sufficient condition for DRL under correlated attributes. Building on this analysis, the paper proposes a new learning framework, SD-HC, designed to disentangle an attribute with multiple modes from other attributes. Empirical results on six datasets demonstrate the effectiveness of SD-HC.

**Strengths:**

1. The motivation of the paper is clear and well-justified.

2. The theoretical analysis in Section 3 is comprehensive and contributes meaningfully to understanding DRL under correlations.

3. The ablation studies are thorough, and the performance of SD-HC is promising.

**Weaknesses:**

1. The discussion on SD-HC framework (Section 4) is confusing. The purpose of batch shuffling and the insight for adopting adversarial training, which is known to be unstable, are not clearly explained. “Shuffling” or “pairing” is indeed a common approach to introduce weak supervision signals in DRL, but it remains unclear whether the shuffling in SD-HC serves a similar role. Moreover, could adversarial training be replaced with a more stable alternative? The motivation for seeking a Nash equilibrium in this context is also not entirely clear.

2. The section on "mode label acquisition" should be clarified, and including an algorithm outline or figure can be helpful for readers to better  understand the procedure. As this component is critical for interpreting the experimental results, its current vague description may easily lead to misunderstandings.

3. More detailed discussion on the insight of variants should be provided in Section 5.3, instead of simple descriptions.

**Questions:**

Please refer to Weaknesses

---

> ### Author Response · Authors · 2025-11-22
> **Response to Reviewer bvj9 (Part 1)**
>
> We sincerely thank the reviewer for recognizing our motivation, theoretical contribution, and experimental thoroughness, and deeply appreciate the constructive and detailed feedback. Below, we address the raised concerns point-by-point and highlight the corresponding improvements made in our paper.
>
> ### **1. The purpose of batch shuffling and adversarial training**
>
> We thank the reviewer for the comment regarding technical details. Specifically, adversarial training and batch shuffling are used to **minimize the conditional mutual information (CMI)** term $I(z_k; z_{-k} \mid m_k)$ for enforcing conditional independence, following the common practice [1,2,3].
>
> * **Equivalence of CMI minimization and joint/marginal representation distribution matching:** Two variables are independent if and only if their joint and marginal distributions match, i.e., for two variables $a,b$, $I(a;b)=0 \Leftrightarrow a \perp b \Leftrightarrow p(a,b)=p(a)p(b)$. Therefore, achieving minimal CMI, $I(z_k; z_{-k} \mid m_k)=0$, is equivalent to matching the distributions, $p(z_k, z_{-k} \mid m_k) = p(z_k \mid m_k)p(z_{-k} \mid m_k)$, where $p(z_k, z_{-k} \mid m_k)$ is the joint distribution of $z_k, z_{-k}$ conditioned on the mode, and $p(z_k \mid m_k)p(z_{-k} \mid m_k)$ is the marginal distributions of $z_k, z_{-k}$ conditioned on the mode.
>
> * **Adversarial training for distribution matching:** Adversarial training with a discriminator is adopted for **matching the joint and marginal distributions** mentioned above. We adopt **Jensen–Shannon divergence** to measure the discrepancy between the joint and marginal distributions, which provides a more **stable** optimization than other divergence measures for mutual information minimization, according to [3].
>
> * **Shuffling for marginal distribution sampling:** The shuffling serves as a **sampling** mechanism from the **marginal distribution** $p(z_k \mid m_k)p(z_{-k} \mid m_k)$ [1,2]. For each mode, sampling from the marginal distribution requires two independent draws from $p(z_k \mid m_k)$ and $p(z_{-k} \mid m_k)$. The $z_k$ in the mini-batch is a draw from $p(z_k \mid m_k)$, and **shuffling** $z_{-k}$ in the mini-batch effectively creates another independent draw from $p(z_{-k} \mid m_k)$. Concatenating the $z_k$ in the order of the mini-batch with the shuffled $z_{-k}$ forms sampled pairs $(z_k, z_{-k})$ from the *marginal distribution*. In contrast, sampled pairs from the *joint distribution* can simply be formed by concatenating $z_k, z_{-k}$ from the same data sample in the mini-batch.
>
> As per the suggestion, we have enriched the explanations in **$\underline{\text{“Losses” (2), Section 4, page 6}}$** of the revised paper to enhance clarity. We hope this clarification helps.
>
> [1] Disentanglement and generalization under correlation shifts, CoLLAs 2022.
>
> [2] Conditional mutual information for disentangled representations in reinforcement learning, NIPS 2023.
>
> [3] Learning deep representations by mutual information estimation and maximization, ICLR 2019.
>
> ---
>
> ### **2. Mode label acquisition process**
>
> We thank the reviewer for the comment regarding the details of the mode label acquisition process. This process is originally described in **$\underline{\text{Algorithm 1, Appendix F, page 27}}$**, and we agree that this algorithm should be referenced in this section to enhance clarity.
>
> Specifically, we first pre-train an encoder for $a_k$ using its supervised attribute prediction loss with attribute labels, then individually perform clustering **under each value of $a_k$** to discover the modes under each attribute value. The modes discovered under different $a_k$ values are indexed together to form mode labels $m_k$.
>
> We have added a reference to this algorithm in **$\underline{\text{“Mode Label Acquisition”, Section 5.1, page 7}}$**. We hope this clarifies the process.

---

> ### Author Response · Authors · 2025-11-22
> **Response to Reviewer bvj9 (Part 2)**
>
> ### **3. Further discussion on the variants**
>
> We appreciate the reviewer’s suggestion to provide deeper insights for variants. We agree that thorough analyses of variants are important for understanding the contribution of each component and for validating the rationality of our design. Below, we provide a detailed discussion regarding the purpose, empirical results, and insights of each variant.
>
> * To investigate the importance of **discrimination loss** and **mode prediction loss**, we remove each of them from **SD‑HC**, yielding variants **SD‑HC‑MP** (without discrimination loss) and **SD‑HC‑MC** (without mode prediction loss). The performance degradation from removing the two losses shows their importance for achieving disentanglement. This is in line with our theoretical results, i.e., enforcing $I(z_k; z_{-k} \mid m_k)=0$ (achieved by discrimination loss) and $I(z_k;m_k)=H(m_k)$ (achieved by mode prediction loss) are essential for yielding disentanglement, as shown in our Proposition 2. Specifically, mode prediction loss makes the model explicitly encode mode information that is important for attribute prediction. Discrimination loss enforces conditional independence between representations, removing redundant information about other attributes for disentanglement.
>
> * To investigate the impact of additional independence constraint, we add an independence constraint conditioned on $a_2$, yielding variant **SD‑HC‑A**. The performance superiority of **SD‑HC** over **SD‑HC‑A** confirms the unnecessity of additional independence constraints, which is consistent with our theoretical results, i.e., one independence constraint conditioned on $m_k$ is sufficient for disentangling $a_k$, as shown in our Proposition 2. Additional independence constraints are unnecessary for disentangling $a_k$, and might hinder training stability, which is probably the cause of performance degradation.
>
> * To investigate the choice of clustering algorithm, we use Marigold [4] for clustering instead of k‑means, yielding the variant **SD‑HC‑MG**. Marigold is a clustering method designed to handle high‑dimensional spaces. The results show that **SD‑HC‑MG** does not outperform **SD‑HC** with k‑means clustering, indicating that **k‑means is sufficient** for our representation scale, and that high-dimensional clustering methods may be unsuitable for this representation scale.
>
> * To investigate the **discriminator architecture**, we design two variants, **SD‑HC‑ID** that uses one *individual discriminator* for each mode, and **SD‑HC‑SD** that uses *one shared discriminator* for all modes, while **SD‑HC** uses one discriminator for each attribute value, *sharing discriminator parameters among the modes under the same attribute value* (see **$\underline{\text{Appendix E, page 27}}$** for detailed architecture). **SD‑HC** consistently outperforms both variants, indicating a good balance in discriminator parameter sharing. In contrast, excessive parameter sharing across all modes may limit the discriminator’s ability to capture differences among modes, whereas no parameter sharing might fail to exploit the commonality shared by the modes under the same attribute value; both might lead to inefficient modeling.
>
> We have included the discussions in **$\underline{\text{Section 5.3, page 8}}$**. We hope these detailed discussions help clarify the rationale behind our design choices and justify the effectiveness of **SD‑HC**.
>
> ---
>
> We sincerely appreciate the reviewer's detailed and constructive comments, which have helped us significantly improve the manuscript. Through the additional discussions and clarifications in this revision, we believe the paper now more comprehensively demonstrates the technical soundness and practical value of our approach. We hope these improvements provide sufficient grounds for a more favorable assessment of our work.

---

### Author Response · Authors · 2025-12-01
**Summary of Revisions**

Dear Area Chairs, Senior Area Chairs, and Program Chairs,

Thank you very much for overseeing the review process and managing our submission. We would like to provide a summary of the rebuttal to offer greater clarity.

Overall, reviewers consistently recognize the novelty, theoretical contribution, and empiral thoroughness of SD-HC, acknowledging it as **"the first formal proof showing sufficiency of conditional mutual information (CMI) minimization"** for the interventional definition of disentanglement, the motivation is **"novel"**, **"clear"**, and **"important"**, the theoretical analysis is **"significant"**, **"rigorous"**, **"contributes meaningfully to understanding DRL under correlations"**, and shows **"a deep understanding of the interplay between attribute and mode information in disentanglement"**, and the empirical evaluation **"demonstrates strong performance across diverse datasets"** and the results are **"promising"** and **"thorough"**.

The reviewers also raised insightful and constructive feedback. We made every effort to address all the concerns by providing detailed clarifications, sufficient evidence, requested results, and in-depth analysis. All the revisions are supplemented in our revised paper and marked in **blue**. Here is the summary of the major revisions:

* ***Practical considerations regarding mode estimation and computational complexity*** (Reviewer uBsu, XBAQ, nRdb):
Mode estimation is addressed from multiple aspects: the robust advantage over baselines under increasing noise levels, additional experiments on a more complex ImageNet‑based dataset, existing sensitivity analyses and weak‑supervision experiments, and our core theoretical contribution and the link to framework design. Regarding computational complexity, we implemented a variant with acceleration techniques, which could better scale to larger datasets.

* ***Conceptual aspects regarding identifiability, causal assumptions, and hierarchical methods*** (Reviewer XBAQ, nRdb):
We clarify that existing hierarchical methods do not consider correlations, thus fail to achieve disentanglement, as validated in the ablation study; our work focuses on disentanglement in an interventional sense, and does not deal with identifiability; and our causal assumptions follow the mainstream of disentanglement works and suit a wide range of real applications.

* ***Empirical discussions and technical details*** (Reviewer bvj9, XBAQ, nRdb): We provided clarifications on details of the training process, explained the choice of disentanglement metrics, and enriched variant discussions.

Thank you again for your thoughtful consideration of our submission. We believe the revisions better highlight the value and scope of our work, demonstrating the significance of our theoretical results and the practical potential of our framework, and hope this summary will assist the Area Chairs in the evaluations.

Best Regards,

Authors

---

### Meta-Review · Area_Chair_far4 · 2025-12-19

**Summary:**

This paper looks at the problem of learning disentangled representations under hidden correlations. They propose a specific generative process, where hidden confounders affect the factors of variation in a specific way, i.e., through a mediating variable related to the multimodality of the distribution, which is then related to the clustering proposed by the authors (which was called out as a limitation in the method by multiple reviewers). This concern was not fully addressed by the authors. It is also related to the question of identifiability (one could probably show that the modes are identifiable with a clusterability assumption and then show identifiability given the mode). The authors commented that the identifiability of the representation is not one of their focuses, citing that their goal is to estimate interventional representations. I believe they fundamentally mistake what it means for a model to be identifiable in causality. Their definition 2 is inherently an identifiability definition in the following sense. On the left-hand side of equation 2 there is a causal estimand involving the do operator. In full generality, this is not computable from data, unless it is *identified* in a statistical estimand (the right-hand side of the equation).

Reviewer XBAQ also asked about the choice of evaluation metric, and the authors argued that they chose mutual information as a better metric for disentanglement in the supervised setting. I found this puzzling frankly. I agree with the authors that evaluation with dependent factors of variations is challenging, which is why I would point them to evaluation metrics in causal representation learning. In particular, they could use conditional independence tests (conditioning on the latent variables that is allegedly disentangled by a representation component, the representation should be independent from other latent variables of repr. dimensions).

**Reviewer Concerns:**

I think the technical questions about the implementation choices (i.e., the framework) that bvj9 were addressed properly. The questions about clustering and identifiability were not addressed sufficiently.

**Reviewer Scores:**

I believe bvj9 should have raised their score based on their initial concerns and the discussion.

---

### Decision · Program_Chairs · 2026-01-26

Reject